# A Policy Optimization Method Towards Optimal-time Stability

**Shengjie Wang**[1,2,3]   **Fengbo Lan**[1]   **Xiang Zheng**[4]

**Yuxue Cao**[1]   **Oluwatosin Oseni**[5]   **Haotian Xu**[1]   **Tao Zhang**[1,†]   **Yang Gao**[1,2,3,†]

[1]Tsinghua University   [2]Shanghai Artificial Intelligence Laboratory
[3]Shanghai Qi Zhi Institute   [4]City University of Hong Kong   [5]Covenant University
† Corresponding author

**Abstract:** In current model-free reinforcement learning (RL) algorithms, stability criteria based on sampling methods are commonly utilized to guide policy optimization. However, these criteria only guarantee the infinite-time convergence of the system's state to an equilibrium point, which leads to sub-optimality of the policy. In this paper, we propose a policy optimization technique incorporating sampling-based Lyapunov stability. Our approach enables the system's state to reach an equilibrium point within an optimal time and maintain stability thereafter, referred to as "*optimal-time stability*". To achieve this, we integrate the optimization method into the Actor-Critic framework, resulting in the development of the Adaptive Lyapunov-based Actor-Critic (ALAC) algorithm. Through evaluations conducted on ten robotic tasks, our approach outperforms previous studies significantly, effectively guiding the system to generate stable patterns.

**Keywords:** Reinforcement Learning, Robotic Control, Stability

## 1   Introduction

Model-free reinforcement learning (RL) controllers have achieved excellent performance in a large variety of robotic tasks [1, 2]. However, current methods lack a stability guarantee, which poses additional risks to both the robots and their environments, especially in the presence of external disturbances [3]. Therefore, ensuring stability in RL-based methods is a crucial requirement.

In control-theoretic methods, there exists an effective tool, Lyapunov functions, to assess the stability [4]. Recently, researchers have employed neural networks to search for feasible Lyapunov functions [5, 6, 7, 8]. Notably, model-free methods have integrated policy and Lyapunov function learning based on discrete sampling-based stability, demonstrating promising results in robotic control tasks [9, 10, 11, 12, 13] (See Appendix A for the discussion of related works). Current research proposes different conditions to speed up the search

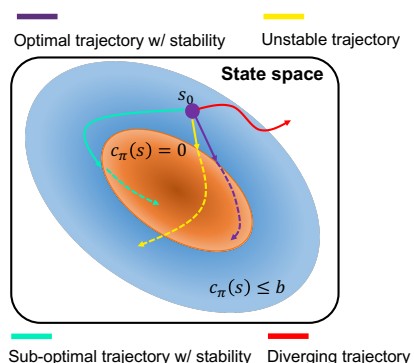

Figure 1: Intuitive example showing the optimality and stability. The stability condition is the mean cost stability defined in Definition 2.1. The figure shows the relationship between three sets, like the whole state space, $\{s \mid c_\pi(s) \leq b\}$ and $\{s \mid c_\pi(s) = 0\}$. The red line represents a diverging trajectory. The yellow line represents a trajectory without stability. The trajectories coloured cyan and purple remain stable, whereas the state in the purple trajectory reaches the set $\{s \mid c_\pi(s) = 0\}$ more quickly.

of a feasible Lyapunov function [9, 10, 11], but these methods only guarantee the eventual convergence of the system's state to an equilibrium point. Figure 1 illustrates a limitation of this guarantee,

7th Conference on Robot Learning (CoRL 2023), Atlanta, USA.

where two stability-maintaining state trajectories (colored cyan and purple) cannot be differentiated by current methods, despite the purple trajectory exhibiting better stability convergence efficiency. In our experiments, we find that this loose guarantee fails to provide adequate guidance for policy learning. Inspired by finite-time stability in continuous-time control theory [14], we propose an improved objective where the state converges to the equilibrium point within an optimal time, which we refer to as "*optimal-time stability*" in this paper. This optimal condition facilitates rapid convergence to the equilibrium point while ensuring stability in the system. Our objective is to develop a policy capable of generating the purple trajectory depicted in Figure 1.

To ensure *optimal-time stability* in the system, we introduce a sampling-based Lyapunov stability certification, which guarantees the fulfillment of the mean cost stability condition. Subsequently, we design a policy optimization method that facilitates the gradual convergence of the policy towards the optimal point, where the system achieves *optimal-time stability*. By leveraging this policy optimization technique, we develop the Adaptive Lyapunov-based Actor-Critic algorithm (ALAC) for learning both a Lyapunov function and policy. Experimental results validate the effectiveness of our approach in ensuring optimality and stability across various robotic control tasks, including legged robot walking and free-floating space robot trajectory planning. Specifically, our method significantly outperforms baselines in ten robotic control environments, and the trained Lyapunov function provides effective guidance for policy learning compared to previous methods[1].

## 2 Problem Formulation

A robotic system can be modelled as a Markov Decision Process (MDP). MDP mainly consists of five elements, $\mathcal{S}$, $\mathcal{A}$, $\mathcal{P}$, $\mathcal{C}$ and $\rho$. Here, $\mathcal{S}$ is the state space, $\mathcal{A}$ is the action space, $\mathcal{P}$ is the dynamic transition function, and $\mathcal{C}$ is the cost function. Besides, the distribution of starting state denotes $s_0 \sim \rho$. At timestep $t$, $s_t \in \mathcal{S}$ represents the state the robot observes. Then, $a_t \in \mathcal{A}$ is the action executed by the agent (robot). Note that $a_t$ is sampled from the agent's policy $\pi(a_t|s_t)$. According to $\mathcal{P}(s_{t+1}|s_t, a_t)$, the state of system transfers to the next state $s_{t+1}$ with a certain probability. At the same time, the agent receives the cost $\mathcal{C}(s_t, a_t)$. And then, we can define the state distribution $\mathcal{T}$:

$$\mathcal{T}(s|\rho, \pi, t+1) = \int_{\mathcal{S}} \int_{\mathcal{A}} \pi(a_t|s_t)\mathcal{P}(s_{t+1}|s_t, a_t)\, \mathrm{d}a\ \mathcal{T}(s|\rho, \pi, t)\, \mathrm{d}s \tag{1}$$

Note that $\mathcal{T}(s|\rho, \pi, 0) = \rho$ holds. An MDP system corresponds to a continuous-state and discrete-time dynamical system with state space $\mathcal{S}$ and action space $\mathcal{A}$. Generally speaking, the system's stability can be verified by a classical tool, Lyapunov's stability function (Appendix B.1). However, the classical definition should be satisfied in the whole state space $\mathcal{S}$, so it is unsuitable for sampling optimization, especially for model-free RL methods. To extend the stability to a reasonable case in model-free RL, we introduce the Mean Cost Stability as the stability condition in this paper.

**Definition 2.1** (Mean Cost Stability [9]). A robotic system remains stable in mean cost when satisfying the following equation, where $c_\pi(s_t) = \mathbb{E}_{a \sim \pi}\mathcal{C}(s_t, a_t)$ and $b$ is a constant.

$$\lim_{t \to \infty} \mathbb{E}_{s_t \sim \mathcal{T}} c_\pi(s_t) = 0, c_\pi(s_0) \leq b \tag{2}$$

Concretely, for most robotic tasks, the mean cost stability is related to the stability of the closed-loop system [9]. Our aim is to reach an equilibrium point within an optimal time and behave stably around it, to achieve the *optimal-time stability*. Thus, we construct the problem formulation represented as follows:

$$\min_{\pi} \mathbb{E}_{\rho, \pi, \mathcal{P}}\Big[\sum_{t=0}^{\infty} \gamma^t c_\pi(s_t)\Big] \qquad s.t. \lim_{t \to \infty} \mathbb{E}_{s_t \sim \mathcal{T}} c_\pi(s_t) = 0 \tag{3}$$

Specifically, the constraint part can facilitate the system's state to converge to an equilibrium point. The objective of the task is to minimize the sum of discounted costs. When the sum of discounted

---

[1]For more information, please visit our project page at https://sites.google.com/view/adaptive-lyapunov-actor-critic.

costs becomes smaller, the system converges to an equilibrium point more quickly. Therefore, by minimizing the sum of discounted costs, we ensure that the system's state converges to an equilibrium point within an optimal time.

## 3 Policy Optimization with Sampling-based Stability

Considering the difficulty of calculating the stability condition in problem (3), we present a novel stability theorem in a sampling-based format suitable for the Reinforcement Learning (RL) setting. Additionally, we transform the original problem (3) into a new optimization problem shown as Equation 8, which progressively seeks the optimal convergence time while ensuring stability.

### 3.1 Sampling-based Stability Certification

Our approach reveals that ensuring the stability condition in Equation (3) corresponds to finding the Lyapunov function $\mathcal{L}$. The Lyapunov function has long been used for stability analysis, and more details can be found in Appendix B.1. Unlike classical methods that rely on the entire state space, we present a certification for sampling-based Lyapunov stability.

**Theorem 3.1** (Sampling-based Lyapunov Stability). *An MDP system is stable with regard to the mean cost as shown in Definition 2.1, if there exists a function $\mathcal{L} : S \to \mathbb{R}$ meets the following conditions:*

$$\alpha c_\pi(s) \leq \mathcal{L}(s) \leq \beta c_\pi(s) \tag{4}$$

$$\mathcal{L}(s) \geq c_\pi(s) + \lambda \mathbb{E}_{s' \sim \mathcal{P}_\pi} \mathcal{L}(s') \tag{5}$$

$$\mathbb{E}_{s \sim \mathcal{U}_\pi}[\mathbb{E}_{s' \sim \mathcal{P}_\pi} \mathcal{L}(s') - \mathcal{L}(s)] \leq -k[\mathbb{E}_{s \sim \mathcal{U}_\pi}[\mathcal{L}(s) - \lambda \mathbb{E}_{s' \sim \mathcal{P}_\pi} \mathcal{L}(s')]] \tag{6}$$

*where $\alpha$, $\beta$, $\lambda$ and $k$ is positive constants. Among them, $\mathcal{U}_\pi = \lim_{T \to \infty} \frac{1}{T} \sum_{t=0}^{T} \mathcal{T}(s \mid \rho, \pi, t)$ represents the stationary distribution of the state, and $\mathcal{P}_\pi(s'|s) = \int_{\mathcal{A}} \pi(a|s)\mathcal{P}(s'|s, a) \, \mathrm{d}a$ represents the stationary distribution of state transition. It is based on some mild assumptions as shown in Appendix C.1. For proof, please see Appendix C.2.*

In practice, the theorem reveals that the guarantee of stability in Equation (3) corresponds to the existence of the Lyapunov function $\mathcal{L}$ under the constraints given by Equations (4), (5), and (6). Intuitively, Equation (4) represents the lower and upper bounds of $\mathcal{L}(s)$. Equation (5) shows the relationship of $\mathcal{L}$ at two steps under the transition distribution. Equation (6) shows the constraint on the expectation of the state stationary distribution. It is worth noting that our method extends the previous approach to a more general case, as illustrated in Appendix C.3. To mitigate the difficulty of searching for $\mathcal{L}$ under multiple constraints, we propose a Lyapunov candidate $\mathcal{L}_\pi(s)$, which naturally meets the constraints (4) and (5), given by $\mathcal{L}_\pi(s) = \mathbb{E}_\pi[\sum_{t=0}^{\infty} \gamma^t c_\pi(s_t)|s_0 = s]$. Please see Appendix B.2 for a detailed demonstration. In summary, the stability condition can be expressed as finding a feasible policy $\pi$, under which the Lyapunov candidate $\mathcal{L}_\pi(s)$ satisfies the constraint in Equation (6).

### 3.2 Policy Optimization Towards Optimal-time Stability

Recalling the optimization problem (3), we observe that the constraint part is equivalent to Equation (6) since the introduction of the Lyapunov candidate $\mathcal{L}_\pi(s)$ eliminates the constraints (4) and (5). Meanwhile, the objective part can be represented as $\min_\pi \mathbb{E}_{s \sim \rho} \mathcal{L}_\pi(s)$ based on the definition of the Lyapunov candidate. Therefore, the optimization problem can be rewritten as:

$$\min_\pi \mathbb{E}_{s \sim \rho} \mathcal{L}_\pi(s)$$
$$\text{s.t. } \mathbb{E}_{s \sim \mathcal{U}_\pi}[\mathbb{E}_{s' \sim \mathcal{P}_\pi} \mathcal{L}_\pi(s') - \mathcal{L}_\pi(s)] \leq -k[\mathbb{E}_{s \sim \mathcal{U}_\pi}[\mathcal{L}_\pi(s) - \lambda \mathbb{E}_{s' \sim \mathcal{P}_\pi} \mathcal{L}_\pi(s')]] \tag{7}$$

A potential approach to tackle the problem (7) is to formulate a unified objective function that removes the constraints. However, directly combining the objective with constraints through simple addition may lead to a sub-optimal policy or an invalid Lyapunov function. Therefore, we propose a policy

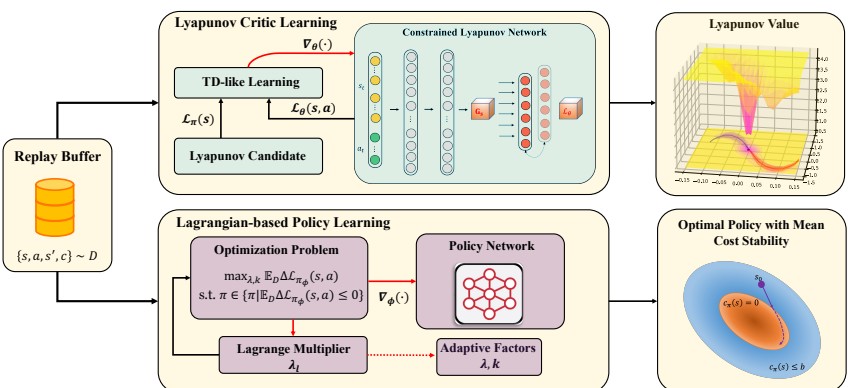

Figure 2: Architecture illustrating the practical implementation of ALAC. The RL agent is optimized by an Actor-Critic framework. The main contents consist of two parts. For the Lyapunov critic learning, the constrained Lyapunov network is updated by the TD-like learning with respect to the Lyapunov candidate. During the policy Learning, we use the Lagrangian-based method to solve the optimization problem shown in Equation (8). Meanwhile, the parameters $\lambda$ and $k$ are adjusted by the Lagrange multiplier $\lambda_l$ adaptively.

optimization method that progressively seeks the optimal policy while ensuring stability by making the constraints' parameters $\lambda$ and $k$ learnable. The method is outlined as follows.

$$\max_{\lambda, k} \mathbb{E}_{s \sim \mathcal{U}_\pi} \Delta \mathcal{L}_\pi(s) \qquad \text{s.t. } \pi \in \{\pi | \mathbb{E}_{s \sim \mathcal{U}_\pi} \Delta \mathcal{L}_\pi(s) \leq 0\} \tag{8}$$

where

$$\Delta \mathcal{L}_\pi(s) = \mathbb{E}_{s' \sim \mathcal{P}_\pi} \mathcal{L}_\pi(s') - \mathcal{L}_\pi(s) + k[\mathcal{L}_\pi(s) - \lambda \mathbb{E}_{s' \sim \mathcal{P}_\pi} \mathcal{L}_\pi(s')] \leq 0 \tag{9}$$

Intuitively, the constraint part corresponds to the constraints defined in Equation (7). The objective seeks to find optimal values for $\lambda$ and $k$ to maximize $\mathbb{E}_{s \sim \mathcal{U}_\pi} \Delta \mathcal{L}_\pi(s)$. Alternatively, we can view this objective as enhancing the strength of the constraints by maximizing $\mathbb{E}_{s \sim \mathcal{U}_\pi} \Delta \mathcal{L}_\pi(s)$. By gradually improving the constraints, minimizing $\mathcal{L}_\pi(s)$ can be achieved in Equation (7). Because, as the constraints become strict, the policy should ensure that $\mathcal{L}_\pi(s')$ at the next state becomes smaller, as dictated by the constraint part in Equation (8). Considering the sampling distribution $\mathcal{U}_\pi$, $\mathcal{L}_\pi$ becomes simultaneously smaller at every state along the state trajectories. This also applies to $\mathcal{L}_\pi$ under the initial state distribution $\rho$. When the strength of the constraints reaches its maximum value, the minimization of $\mathcal{L}_\pi(s)$ is achieved. In addition, we offer a comprehensive illustration to explain the decrease of $\mathcal{L}_\pi(s')$ using a classical tracking task, which is provided in Appendix C.5. Interestingly, the form of Equation (9) bears resemblance to the finite-time tracking method in continuous-time systems (Appendix C.4).

## 4 Adaptive Lyapunov-based Actor-Critic Algorithm

In this section, we propose the Adaptive Lyapunov-based Actor-Critic algorithm (ALAC) to solve the optimization problem (8). According to the variables shown in Equation (8), we introduce two learnable parameters $\lambda$ and $k$, along with two neural networks, namely the Lyapunov critic $\mathcal{L}_\theta(s, a)$ and the actor $\pi_\phi(a|s)$. The Lyapunov critic $\mathcal{L}_\theta(s, a)$ serves a dual role in both stability analysis and actor learning, and it depends on both the state $s$ and the action $a$. We leverage the relationship $\mathbb{E}_\pi \mathcal{L}_\theta(s, a) = \mathcal{L}_\pi(s)$ to compute $\mathcal{L}_\pi(s)$ in Equation (8). Furthermore, the actor $\pi_\phi(a|s)$ maps a given state $s$ to a distribution over actions. This action distribution is modeled as a Gaussian distribution with a state-dependent mean $\mu_\phi(s)$ and a diagonal covariance matrix $\Sigma_\phi(s)$.

Figure 2 shows the architecture of the ALAC algorithm. Specifically, we employ the Lagrangian-based method to handle the solution of the policy under the constraint in (8). Additionally, the parameters $\lambda$ and $k$ are adjusted by the Lagrange multiplier $\lambda_l$ to satisfy the objective in (8).

### 4.1 Lyapunov Critic Learning

For the training of $\mathcal{L}_\theta(s, a)$, we typically update $\theta$ based on the TD error in an actor-critic learning process. In our framework, the target Lyapunov value can be approximated as $c_\pi + \gamma \mathcal{L}'(s', \pi'(\cdot|s'))$. Thus, the update rule for $\theta$ can be written as:

$$\theta_{k+1} = \theta_k + \alpha_\theta(\nabla_\theta(\mathcal{L}_\theta(s, a) - (c_\pi + \gamma \mathcal{L}'(s', \pi'(\cdot|s'))))^2) \tag{10}$$

where $k$ is the number of iterations. $\mathcal{L}'$ and $\pi'$ are the target networks parameterized by $\theta'$ and $\phi'$, respectively. To ensure stable learning, the parameters $\theta'$ and $\phi'$ are typically updated using an exponentially moving average of weights, controlled by a hyper-parameter $\sigma \in (0, 1)$. In order to encourage accurate and efficient learning, we construct a constrained critic network. The constrained network ensures that the output is non-negative and the Lyapunov value should be zero when the state is an equilibrium point. For the details, please see Appendix C.6.

### 4.2 Lagrangian-based Policy Learning

Policy learning aims to search feasible parameters of $\lambda$, $k$ and the policy network to solve the optimization problem defined in Equation (8). According to Equation (9) and Lyapunov critic, we denote $\Delta \mathcal{L}_{\pi_\phi}(s, a)$ as $\mathcal{L}_\theta(s', \pi_\phi(\cdot \mid s')) - \mathcal{L}_\theta(s, a) + k[\mathcal{L}_\theta(s, a) - \lambda \mathcal{L}_\theta(s', \pi_\phi(\cdot \mid s'))]$.

First, we solve the sub-problem of Equation (8), finding $\pi_\phi$ when satisfying the constraint with arbitrary $\lambda$. Applying the Lagrangian-based method [15], the parameters of $\pi_\phi$ are updated by:

$$\phi_{k+1} = \phi_k + \alpha_\phi(\lambda_l \nabla_a \Delta \mathcal{L}_{\pi_\phi}(s, a) \nabla_\phi \pi_\phi(s, a)) \tag{11}$$

where $\alpha_\phi$ is the learning rate of $\phi$. $\lambda_l$ represents the Lagrange multiplier of the constraint. During the training, $\lambda_l$ is updated by gradient ascent to maximize $\Delta \mathcal{L}_{\pi_\phi}(s, a)$.

$$\lambda_l^{k+1} = \lambda_l^k + \alpha_{\lambda_l} \Delta \mathcal{L}_{\pi_\phi}(s, a) \tag{12}$$

Note that $\lambda_l$ should always be positive. $\alpha_{\lambda_l}$ is the learning rate. It is worth noting that $\lambda_l$ is clipped by 0 and 1 to bound the value, which is exploited in optimization methods [16].

According to Equation (8), we need to find suitable values for $\lambda$ and $k$ to maximize $\Delta \mathcal{L}_{\pi_\phi}(s, a)$. We observe that $\lambda$ should decrease towards 0 to enlarge $\Delta \mathcal{L}_{\pi_\phi}(s, a)$. Furthermore, it should be between $\gamma$ and 0, with the range of $\lambda$ restricted based on Theorem 3.1. Enlarging $\Delta \mathcal{L}_{\pi_\phi}(s, a)$ corresponds to strengthening the constraints in Equation (8). Notably, we find that the Lagrange multiplier $\lambda_l$ decreases as the constraints are satisfied. Thus, we use the rule $\lambda \leftarrow \min(\lambda_l, \gamma)$ to gradually improve the constraints. As for the selection of $k$, we adjust its value based on the Lagrange multiplier, setting $k \leftarrow 1 - \lambda_l$. By increasing $k$, the strength of the constraints gradually improves throughout the training process. Finally, when $\lambda$ and $k$ stabilize, indicating that the Lagrange multiplier $\lambda_l$ remains constant, we can observe that $\Delta \mathcal{L}_{\pi_\phi}(s, a)$ approaches 0 according to Equation (12). This implies that the objective of maximizing $\Delta \mathcal{L}_{\pi_\phi}(s, a)$ is achieved since the maximum value of $\Delta \mathcal{L}_{\pi_\phi}(s, a)$ does not exceed 0 due to the constraint part in Equation (8).

In addition, to improve the exploration efficiency, we add a constraint about the minimum entropy as the same as the maximum entropy RL algorithms. Until now, we have designed the complete ALAC algorithm, and the pseudo-code is provided in Appendix D.

### 4.3 Theoretical Analysis

Theorem 3.1 has assumed that the expectation is obtained perfectly, but this is not the case in practical settings due to finite samplings. Thus, we derive the bias between the practical computing and theoretical analysis about $\mathcal{U}_\pi$.

To be concrete, we need an infinite number of trajectories with infinite time steps to estimate the distribution $\mathcal{U}_\pi$. Whereas in practice, only $M$ trajectories of $T$ time steps are accessible. To better illustrate the issue, we define a finite sampling distribution $\mathcal{U}_\pi^T$, apparently where $\mathcal{U}_\pi^T = \frac{1}{T} \sum_{t=0}^{T} \mathcal{T}(s \mid \rho, \pi, t)$. First, we provide a quantitative bound of expectation from $\mathcal{U}_\pi$ and $\mathcal{U}_\pi^T$.

Table 1: Performance evaluations of the cultivated costs and stability constraint violations on ten environments compared with six baselines. All quantities are provided in a scale of $0.1$. Standard errors are provided in brackets. (if the mean constraint violations are less than 0.2, the sign is ↓ else ↑.'-' indicates the algorithm does not contain the stability constraints.)

| Task | Metrics | ALAC | SAC-cost | SPPO | LAC | LAC* | POLYC | LBPO | TNLF |
|---|---|---|---|---|---|---|---|---|---|
| Cartpole-cost | Cost Return | 26.2(7.0) | **22.7**(12.6) | 102.3(59.3) | 31.0(10.1) | 31.5(5.1) | 104.8(70.7) | 205.3(27.0) | 33.5(24.5) |
| | Violation | ↓ | - | ↑ | ↓ | ↓ | ↓ | - | ↓ |
| Point-circle-cost | Cost Return | **111.1**(4.5) | 111.8(2.4) | 247.9(58.2) | 958.6(15.5) | 112.0(5.0) | 207.0(62.4) | 722.1(126.1) | 145.8(38.0) |
| | Violation | ↓ | - | ↑ | ↓ | ↑ | ↑ | - | ↓ |
| Halfcheetah-cost | Cost Return | **1.7**(0.7) | 16.6(25.2) | 144.0(14.6) | 119.5(37.3) | 1.8(0.5) | 168.8(10.7) | 37.8(24.8) | 6.5(1.4) |
| | Violation | ↓ | - | ↑ | ↓ | ↓ | ↓ | - | ↓ |
| Swimmer-cost | Cost Return | **44.6**(4.8) | 53.7(12.4) | 52.5(4.2) | 47.5(1.3) | 44.8(3.0) | 104.7(11.0) | 52.3(11.3) | 46.5(2.4) |
| | Violation | ↓ | - | ↑ | ↓ | ↑ | ↑ | - | ↓ |
| Ant-cost | Cost Return | **101.0**(42.1) | 155.2(29.9) | 255.0(31.2) | 166.9(13.6) | 125.6(12.5) | 259.8(37.1) | 114.6(26.1) | 186.8(11.0) |
| | Violation | ↓ | - | ↑ | ↓ | ↑ | ↑ | - | ↓ |
| Humanoid-cost | Cost Return | 354.6(97.1) | 441.9(18.3) | 531.8(22.9) | 431.3(14.9) | 368.3(76.6) | 490.4(32.5) | 452.4(13.9) | **317.7**(31.1) |
| | Violation | ↓ | - | ↑ | ↓ | ↑ | ↑ | - | ↓ |
| Minitaur-cost | Cost Return | 493.0(67.9) | 692.2(93.0) | 950.0(72.3) | 612.2(47.8) | 666.6(306.7) | 608.3(65.6) | 838.3(237.0) | **382.9**(62.6) |
| | Violation | ↓ | - | ↑ | ↓ | ↑ | ↑ | - | ↓ |
| Spacereach-cost | Cost Return | **1.6**(0.2) | 8.9(8.8) | 19.4(2.5) | 35.2(1.6) | 1.8(0.4) | 125.7(20.8) | 31.0(19.1) | 112.1(53.0) |
| | Violation | ↓ | - | ↓ | ↓ | ↓ | ↓ | - | ↓ |
| Spacerandom-cost | Cost Return | **2.3**(0.3) | 38.4(28.6) | 53.2(32.7) | 33.9(3.5) | 2.8(0.9) | 112.8(19.4) | 35.8(2.9) | 85.9(42.3) |
| | Violation | ↓ | - | ↓ | ↓ | ↓ | ↓ | - | ↓ |
| Spacedualarm-cost | Cost Return | **26.1**(3.5) | 36.1(8.3) | 201.9(48.8) | 66.3(10.6) | 63.6(62.1) | 140.6(17.4) | 37.8(7.5) | 280.1(99.3) |
| | Violation | ↓ | - | ↓ | ↓ | ↓ | ↓ | - | ↓ |

**Theorem 4.1.** *Suppose that the length of sampling trajectories is $T$, then the bound can be expressed as:*

$$|\mathbb{E}_{s\sim\mathcal{U}_\pi}\Delta\mathcal{L}_\pi(s) - \mathbb{E}_{s\sim\mathcal{U}_\pi^T}\Delta\mathcal{L}_\pi(s)| \leq 2\frac{(k+1)\overline{c_\pi}}{1-\gamma}T^{q-1} \tag{13}$$

*where $\overline{c_\pi}$ is the maximum of cost and $q$ is a constant in $(0,1)$. For proof, please see Appendix C.7.*

Next, we take the number of trajectories into consideration and derive the probabilistic bound of the difference of $\Delta\mathcal{L}_\pi(s)$ estimated by $\mathcal{U}_\pi^T$ distribution and $M$ trajectories.

**Theorem 4.2.** *Suppose that the length of sampling trajectories is $T$ and the number of trajectories is $M$, then there exists the following upper bound:*

$$\mathbb{P}(|\frac{1}{MT}\sum_{m=1}^{M}\sum_{t=1}^{T}\Delta\mathcal{L}_\pi(s_t^m) - \mathbb{E}_{s\sim\mathcal{U}_\pi^T}\Delta\mathcal{L}_\pi(s)| \geq \alpha) \leq 2\exp(-\frac{M\alpha^2(1-\gamma)^2}{((1-k\lambda)^2 + (k-1)^2)\overline{c_\pi}^2}) \tag{14}$$

*where $s_t^m$ represents the state in the $m$-th trajectory at the time $t$. For proof, please see Appendix C.8.*

Theorems 4.1 and 4.2 highlight the theoretical gap between infinite and finite samples in practical usage. In addition, they provide valuable insights into the choice of $k$. Theorem 4.1 suggests that when $k$ approaches 0, the gap becomes small in practice. On the other hand, Theorem 4.2 indicates that $k$ is better to be set to 1. This implies that the optimal choice of $k$ lies within the range of 0 to 1.

## 5 Experiments

In this section, we demonstrate empirical evidence that **ALAC** captures an improved trade-off between optimality (sum of costs) and stability compared to the baseline approaches. We test our method and baselines in ten robotic control environments. Details of ten environments are given in Appendix E.1. Furthermore, we benchmark the **ALAC** method against five algorithms with a neural Lyapunov function, including **POLYC** [10], **LBPO**[17], **TNLF**[11], **SPPO**[18] and **LAC** [9] [2]. We also take the **SAC-cost** [19] method into account because the method is very close to our method without the stability condition. For the detailed hyper-parameter settings see Appendix E.2.1 and E.2.2.

---

[2]We find **LAC** with a large $\alpha_3$ (see Appendix E.2.1) performs better, so we call it **LAC**\* for the distinction between them.

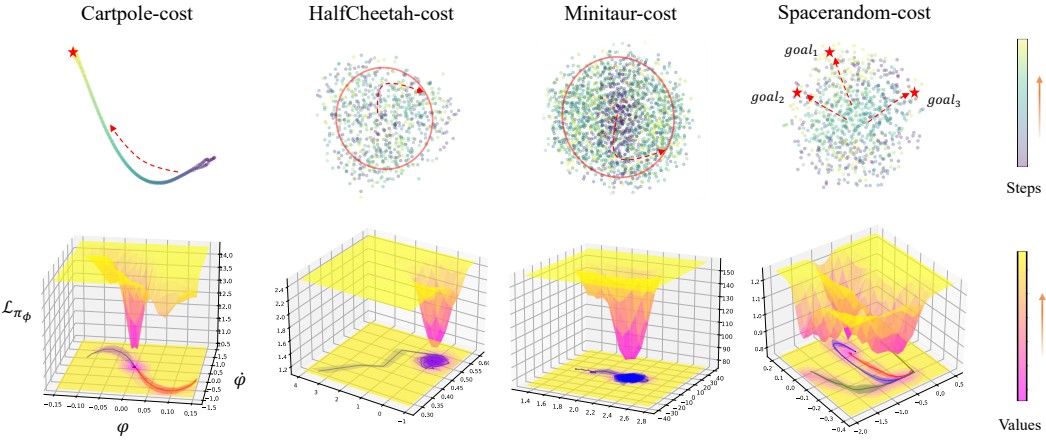

Figure 3: Visualization of states for ALAC method by t-SNE and phase trajectory techniques. The top row of the figure depicts the t-SNE dimension reduction technique. (**Cartpole-Cost** is visualized within 2 dims while others within 3 dims.) The bottom row shows the phase trajectories and Lyapunov-value surfaces. $\psi$ and $\dot{\psi}$ denotes the angular position and velocity respectively.

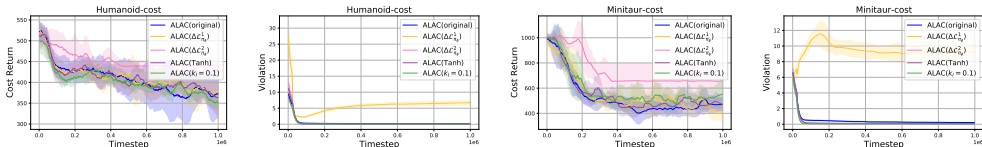

Figure 4: Ablation studies of the sampling-base stability we propose. ALAC(original) shows comparable or the best performance compared with other certifications on each task.

## 5.1 Comparing with Baselines

In this part, we evaluate the optimality and stability of our methods and baselines. According to the definition of *optimal-time stability* in Eq. (3), we use the accumulated cost as the metric of optimality and the stability constraint violations as the stability metric in a testing episode (further details can be found in Appendix E.3). The results demonstrate that **ALAC** achieves the lowest accumulated cost and stability violations. To fairly evaluate the performance of the methods mentioned above, we run the experiments over 5 rollouts and 5 seeds for all algorithms. Table 1 shows the performance on all tasks, and the training curves for different algorithms are in Appendix E.3. Although **LAC**$^*$ using tighter constraints achieves comparable performance with our method in contrast to **LAC**, the stability violations in **LAC**$^*$ remain at a high level on many tasks. Admittedly, **TNLF** achieves lower cost than **ALAC** on **Minitaur-cost**, but **TNLF** converges to suboptimal policies on many tasks. According to Figure 10, we notice that in **TNLF** the trained Lyapunov function is close to 0 quickly. Hence, it does not provide dense guidance for the policy, thus leading it to a suboptimal solution.

## 5.2 Ablation Studies

To demonstrate the effectiveness of $\Delta\mathcal{L}_{\pi_\phi} \leq 0$, we do the ablation studies about different constraints in **ALAC**. We compare the performance of the original **ALAC** with a version that uses $\Delta\mathcal{L}^1_{\pi_\phi}$ ($\lambda = 0$) and $\Delta\mathcal{L}^2_{\pi_\phi}$ ($\lambda = 1$), and with a version where $k$ is a constant throughout training. The descriptions of ablation algorithms are given in Appendix E.4. Figure 4 and Figure 9 (see Appendix D.4) depict the accumulated cost and constraint violations on all tasks. **ALAC**($\Delta\mathcal{L}^2_{\pi_\phi}$) achieve lower performance than **ALAC**, while **ALAC**($\Delta\mathcal{L}^1_{\pi_\phi}$) performs the tasks comparably with **ALAC**. Nevertheless, more

Table 2: Average evaluation score and standard deviation on our environments for ALAC with and without the errors under different biases of goals. (w/ errors means using errors between the desired and achieved goals as extra elements of states for the agent)

| Task | Point-circle-cost | | | Halfcheetah-cost | | | Spacereach-cost | | |
|---|---|---|---|---|---|---|---|---|---|
| Biases of goals | -20% | 0% | 20% | -20% | 0% | 20% | -20% | 0% | 20% |
| ALAC w/ errors | 85.2$_{(4.5)}$ | 110.1$_{(3.9)}$ | 148.5$_{(12.2)}$ | **3.9**$_{(0.8)}$ | **2.5**$_{(0.6)}$ | **8.3**$_{(4.7)}$ | **6.4**$_{(1.7)}$ | 2.4$_{(1.4)}$ | **8.7**$_{(1.7)}$ |
| ALAC w/o errors | 178.8$_{(7.8)}$ | 118.9$_{(11.4)}$ | 247.8$_{(11.9)}$ | 10.1$_{(2.1)}$ | 3.3$_{(1.2)}$ | 13.4$_{(2.1)}$ | 11.9$_{(0.4)}$ | **1.6**$_{(0.2)}$ | 11.5$_{(0.3)}$ |
| SAC-cost w/ errors | **84.2**$_{(4.2)}$ | **109.3**$_{(2.2)}$ | **140.1**$_{(3.0)}$ | 60.1$_{(27.5)}$ | 81.6$_{(50.2)}$ | 129.5$_{(85.6)}$ | 21.9$_{(12.3)}$ | 22.1$_{(16.9)}$ | 20.6$_{(18.1)}$ |
| SAC-cost w/o errors | 180.9$_{(6.3)}$ | 115.3$_{(4.0)}$ | 240.3$_{(3.7)}$ | 15.9$_{(15.7)}$ | 16.7$_{(25.5)}$ | 33.5$_{(34.0)}$ | 16.1$_{(6.2)}$ | 8.8$_{(8.8)}$ | 15.0$_{(7.2)}$ |

strict constraints ($\mathbf{ALAC}(\Delta\mathcal{L}_{\pi_\phi}^1)$) negatively affect its performance on constraint violations, as shown in Figure 4. This is because there doesn't exist a reasonable policy that meets such strict constraints. Moreover, the results of $\mathbf{ALAC}(k = 0.1)$ comparing with **ALAC** demonstrate that the heuristic updating of $k$ is effective during the training.

## 5.3 Evaluation Results

**Analysis of Visualization**: First, we use the t-SNE method to indicate the visualization of the state in 3 dimensions in order to illustrate better the stability of the system learned by **ALAC** (**Cartpole-cost** in 2 dimensions). As we can see, the top row of Figure 3 shows the states in the final stage of an episode converge to a point or circle. Basically, we recognize that those patterns happen in a stable system. The second row of Figure 3 shows the phase trajectories with variance according to the state pairs of joint angular position and velocity. In practice, experts can judge a system's stability from a phase space of the system. Concretely, $\psi$ and $\dot{\psi}$ represent an angular position and velocity, respectively. The angular velocity starts from 0 to 0, and the angular position starts from the beginning to an equilibrium point. Based on the above phenomenons, it suggests the trained systems using the **ALAC** method satisfy focal stability or stable limit cycles. Furthermore, the Lyapunov value exhibits significant changes in the state space, as depicted in Figure 3 (bottom row). It indicates that the Lyapunov value can effectively guide the policy towards discovering stable patterns in the system. More implementation details for t-SNE and phase trajectories are given in Appendix E.5.

**Generalization**: Furthermore, some experiments verify that the policy can generalize well to follow previously unseen desired goals, such as different desired velocities in **Halfcheetah-cost**. When evaluating, we add $\pm 20\%$ bias to the desired goals in three environments, as illustrated in Table 2. The definition of desired and achieved goals is shown in Appendix E.6.2. Specifically, when we introduce the error between the desired and achieved goals as additional information in the state, **ALAC w/ errors** gains remarkable performance improvement on generalization compared to **SAC-cost w/ errors**. This indicates that the Lyapunov function successfully captures the relationship between error and stability. Even though the desired goal experiences some shifting, the Lyapunov function can guide the policy to reach the goal. Finally, we also add external disturbances with different magnitudes in ten environments and observe the performance difference. Figure 8 (see Appendix E.6.1) shows that in all scenarios, **ALAC** enjoys superior performance over other methods.

## 6 Discussion

**Conclusion.** We propose a sampling-based Lyapunov stability condition to meet the mean cost stability. Based on the condition, the policy optimization with sampling-based stability is proposed to gradually find the optimal policy which maintains the *optimal-time stability* we propose. Based on the Actor-Critic framework and Lagrangian-based optimization, We present a practical algorithm, namely the Adaptive Lyapunov-based Actor-Critic algorithm (ALAC).

**Limitation.** Despite the great success in simulated tasks, our method has not been evaluated in practical scenarios. An important direction for future work is to apply our method to real-world robotic control tasks, such as locomotion and navigation. While this work primarily focuses on providing stability guarantees, it is essential to consider another critical requirement, safety.

**Acknowledgments**

This work is supported by the Ministry of Science and Technology of the People´s Republic of China, the 2030 Innovation Megaprojects "Program on New Generation Artificial Intelligence" (Grant No. 2021AAA0150000). This work is also supported by the National Key R&D Program of China (2022ZD0161700).

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

# A   Related Work

Learning-based controllers have achieved excellent performance in non-linear dynamic systems [1, 2]. However, a lack of stability introduces additional risks to the agents and environments [3]. Fortunately, there exists an effective tool, Lyapunov functions, to assess the stability. Lyapunov functions can be designed for a linear system with specific criteria in the form of a quadratic positive-definite function. But how to find a suitable Lyapunov function remains an open challenge in the non-linear dynamic system [4].

## A.1   Model-based RL & Lyapunov Learning

Due to the difficulty of a manual design, constructing a Lyapunov neural network has become increasingly popular in a non-linear dynamic system. For the model-known situation, the approaches jointly learn a Lyapunov function and a controller [20, 21, 22, 23, 24, 25, 7]. But it restricts the application of complex systems which are hard to obtain accurate models. Therefore, some researchers present model-learned methods with a stability guarantee, in which Gaussian Process or Neural Network approximates the model. The model-learned method can be separated into two types. The first one is learning dynamics models guided by a learnable Lyapunov function, in which policies are inherently included or learned by LQR method [26, 8, 27, 28, 29]. Another approach is to construct a learnable policy network updated by a neural Lyapunov function, thereby satisfying the stability of system [30, 31, 8, 5, 6]. However, we notice that most model-learned methods are only verified in relatively easy environments. A possible reason is that the coupling of the Lyapunov function and dynamic model makes learning unstable or incompatible due to interdependency.

## A.2   Model-free RL & Lyapunov Learning

A promising direction is to study model-free methods with a stability guarantee. Recently, a large variety of methods have been proposed to address the issue. One method is that the policy is updated by a mixed objective with respect to the neural Lyapunov function and Q function. POLYC [10] introduced the necessary conditions required by the Lyapunov function into objectives to optimize the policy network. LBPO [17] applied the logarithmic barrier function based on the form of the Lyapunov function. TNLF [11] constructed Lyapunov V and Q functions trained by the stability certification. The other form is policy optimization with Lyapunov constraints. Chow et al. [32] designed a constrained RL algorithm to project a policy in a trust region with Lyapunov stability. In those previous study, there still exist a main drawback. The discrete Lyapunov condition they used did not meet the demand for a sampling-based stability guarantee in RL. Therefore, Han et al. [9] considered a sampling-based stability condition they proposed as a constraint and then used the prime-dual method to modify the constraint. Their latter work verified them in both on-policy and off-policy settings [13, 33]. Nevertheless, their method can only find an existing policy with the demand for stability. In contrast, our method can satisfy the sampling-based stability and search for the optimal policy adaptively. The optimal policy can facilitate the state to reach the equilibrium point rapidly.

## A.3   Applications

Furthermore, it is worth noting that current RL-based methods with stability guarantee have been applied in some practical problems successfully, such as monitoring the security of interconnected microgrids [34], power system control [12], automatic assembly [35] and motion planning of autonomous vehicles [36].

# B  Preliminary Remarks

## B.1  Lyapunov function

The method we employ is constructing or finding a Lyapunov function, denoted as $\mathcal{L}: \mathcal{S} \to \mathbb{R}$, with the property that its difference along the state trajectory is negative definite. This ensures that the state moves in a direction that decreases the value of the Lyapunov function, eventually leading to convergence at the origin. The Lyapunov method has a well-established history of application in stability analysis and controller design within control theory.

**Definition B.1** (Equilibrium Point). A state $s_e$ is an equilibrium point if $\exists$ action $a_e \in \mathcal{A}$ such that $f(s_e, a_e) = s_e$. [37]

**Definition B.2** (Stabilizable in the sense of Lyapunov). A system is stabilizable if $\forall \epsilon > 0, \exists \delta$ such that for all $s_0 \in \mathcal{S}$ such that $||s_0 - s_e|| \leq \delta$, there exists $\{a_t\}_{t=0}^{\infty}$ such that the resulting $\{s_t\}_{t=0}^{\infty}$ satisfies $||s_t - s_e|| \leq \epsilon \, \forall t \geq 0$.[37]

**Definition B.3** (Lyapunov Function). A continuous and radially unbounded function $\mathcal{L}: \mathcal{S} \to \mathbb{R}$ is a Lyapunov function if the following conditions hold:

1. $\forall s \in S, \exists a \in \mathcal{A}$, s.t. $\mathcal{L}(s) \geq \mathcal{L}(f(s, a))$,

2. $\forall s \neq 0, \mathcal{L} > 0; \mathcal{L}(0) = 0$.

If a Lyapunov function exists, a discrete-time system can achieve stability in the sense of Lyapunov without considering the physical energy.

## B.2  Lyapunov Candidate Bound

In this part, we show that the Lyapunov candidate $\mathcal{L}_\pi(s)$ meets the property in Theorem 3.1, which can be formulated as:

$$c_\pi(s) + \lambda \mathbb{E}_{s' \sim \mathcal{P}_\pi} \mathcal{L}(s') \leq \mathcal{L}(s) \leq \beta c_\pi(s) \tag{15}$$

where we omit the lower bound $\alpha c_\pi(s)$ which is naturally satisfied by $\mathcal{L}_\pi(s)$.

Firstly, according to the definition of $\mathcal{L}_\pi(s)$, we have

$$
\begin{aligned}
\mathcal{L}_\pi(s) &= \mathbb{E}_\pi[\sum_{t=0}^{\infty} \gamma^t c_\pi(s_t)|s_0 = s] \\
&= \mathbb{E}_\pi[c_\pi(s_0) + \sum_{t=1}^{\infty} \gamma^t c_\pi(s_t)|s_0 = s] \\
&= c_\pi(s) + \mathbb{E}_\pi[\sum_{t=1}^{\infty} \gamma^t c_\pi(s_t)] \\
&= c_\pi(s) + \gamma \mathbb{E}_{\pi,s' \sim \mathcal{P}_\pi}[\sum_{t=0}^{\infty} \gamma^t c_\pi(s_t)|s_0 = s'] \\
&= c_\pi(s) + \gamma E_{s' \sim \mathcal{P}_\pi} \mathcal{L}(s')
\end{aligned}
\tag{16}
$$

Considering the left-hand side of Equation (4), we can find that if $\lambda \leq \gamma$ holds, the lower bound of the Lyapunov function can be satisfied. This is because the Lyapunov candidate $\mathcal{L}_\pi(s)$ is positive at each state. Furthermore, the right-hand side of Equation 4 illustrates the higher bound of the Lyapunov function exists. The condition is also guaranteed for our Lyapunov candidate shown in the following

process.

$$\mathcal{L}_\pi(s) = \mathbb{E}_\pi[\sum_{t=0}^\infty \gamma^t c_\pi(s_t)|s_0 = s]$$

$$\leq \sum_{t=0}^\infty \gamma^t \mathbb{E}_\pi[c_\pi(s_t)|s_0 = s] \tag{17}$$

$$\leq \frac{\overline{c_\pi}}{1-\gamma}$$

Note that $\overline{c_\pi}$ denotes the maximum cost. The second row of the inequality holds due to Jensen inequality. Only if the maximum cost exists, $\exists \beta \in \mathbb{R}_+, \frac{\overline{c_\pi}}{1-\gamma} \leq \beta c_\pi(s)$ holds.

## C  Details of Theoretical Analysis

### C.1  Assumptions of Theorem 3.1

**Assumption C.1** (Region of Attraction). There exists a positive constant $b$ such that $\rho(s) > 0, \forall s \in \{s|c_\pi(s) \leq b\}$.

**Assumption C.2** (Ergodic Property). The Markov Chain driven by the policy $\pi$ is ergodic, $\omega_\pi(s) = \lim_{t\to\infty} \mathcal{T}(s|\rho, \pi, t)$.

The first one ensures that the starting state is sampled in the region of attraction. The second one is the existence of the stationary state distribution.

### C.2  Proof of Theorem 3.1

**Theorem C.3** (Sampling-based Lyapunov Stability). *An MDP system is stable with regard to the mean cost, if there exists a function $\mathcal{L} : S \to \mathbb{R}$ meets the following conditions:*

$$\alpha c_\pi(s) \leq \mathcal{L}(s) \leq \beta c_\pi(s) \tag{18}$$

$$\mathcal{L}(s) \geq c_\pi(s) + \lambda \mathbb{E}_{s'\sim\mathcal{P}_\pi}\mathcal{L}(s') \tag{19}$$

$$\mathbb{E}_{s\sim\mathcal{U}_\pi}[\mathbb{E}_{s'\sim\mathcal{P}_\pi}\mathcal{L}(s') - \mathcal{L}(s)] \leq -k[\mathbb{E}_{s\sim\mathcal{U}_\pi}[\mathcal{L}(s) - \lambda\mathbb{E}_{s'\sim\mathcal{P}_\pi}\mathcal{L}(s')]] \tag{20}$$

*where $\alpha$, $\beta$, $\lambda$ and $k$ is positive constants. Among them, $\mathcal{U}_\pi = \lim_{T\to\infty} \frac{1}{T}\sum_{t=0}^T \mathcal{T}(s \mid \rho, \pi, t)$ represents the stationary distribution of the state, and $\mathcal{P}_\pi(s'|s) = \int_\mathcal{A} \pi(a|s)\mathcal{P}(s'|s,a)\,\mathrm{d}a$ represents the stationary distribution of state transition.*

*Proof.* Firstly, we simplify the left side of the Equation (20) with reference to [9]. Introducing the definition of $\mathcal{U}_\pi(s)$ leads to

$$\mathbb{E}_{s\sim\mathcal{U}_\pi}[\mathbb{E}_{s'\sim\mathcal{P}_\pi}\mathcal{L}_\pi(s') - \mathcal{L}_\pi(s)]$$

$$= \int_\mathcal{S} \lim_{T\to\infty} \frac{1}{T}\sum_{t=0}^T \mathcal{T}(s \mid \rho, \pi, t)(\int_\mathcal{S} \mathcal{P}_\pi(s'|s)\mathcal{L}_\pi(s')\mathrm{d}s' - \mathcal{L}_\pi(s))\mathrm{d}s \tag{21}$$

Due to the boundedness of $\mathcal{L}_\pi$, we apply the Lebesgue's Dominated convergence theorem. To be specific, when $|F_n(s)| \leq B(s), \forall s \in \mathcal{S}, \forall n$ holds, we have

$$\lim_{n\to\infty} \int_\mathcal{S} F_n(s)\mathrm{d}s = \int_\mathcal{S} \lim_{n\to\infty} F_n(s)\mathrm{d}s \tag{22}$$

Hence, we get

$$\mathbb{E}_{s\sim\mathcal{U}_\pi}[\mathbb{E}_{s'\sim\mathcal{P}_\pi}\mathcal{L}_\pi(s') - \mathcal{L}_\pi(s)]$$

$$= \int_{\mathcal{S}} \lim_{T\to\infty} \frac{1}{T} \sum_{t=0}^{T} \mathcal{T}(s \mid \rho, \pi, t)(\int_{\mathcal{S}} \mathcal{P}_\pi(s'|s)\mathcal{L}_\pi(s')\mathrm{d}s' - \mathcal{L}_\pi(s))\mathrm{d}s$$

$$= \lim_{T\to\infty} \int_{\mathcal{S}} \frac{1}{T} \sum_{t=0}^{T} \mathcal{T}(s \mid \rho, \pi, t)(\int_{\mathcal{S}} \mathcal{P}_\pi(s'|s)\mathcal{L}_\pi(s')\mathrm{d}s' - \mathcal{L}_\pi(s))\mathrm{d}s \qquad (23)$$

$$= \lim_{T\to\infty} \frac{1}{T}(\sum_{t=1}^{T+1} \mathbb{E}_{\mathcal{T}(s|\rho,\pi,t)}\mathcal{L}_\pi(s) - \sum_{t=0}^{T} \mathbb{E}_{\mathcal{T}(s|\rho,\pi,t)}\mathcal{L}_\pi(s))$$

$$= \lim_{T\to\infty} \frac{1}{T}(\mathbb{E}_{\mathcal{T}(s|\rho,\pi,T+1)}\mathcal{L}_\pi(s) - \mathbb{E}_{\mathcal{T}(s|\rho,\pi,t=0)}\mathcal{L}_\pi(s))$$

Note that $\mathcal{T}(s|\rho,\pi,t=0)$ is equal to $\rho$. Since the expectation of $\mathcal{L}_\pi(s)$ is a finite value, the left side of Equation (20) is zero.

Now, we turn to the right side of Equation (20). According to the Equation (23), we have

$$-k[\mathbb{E}_{s\sim\mathcal{U}_\pi}[\mathcal{L}(s) - \lambda\mathbb{E}_{s'\sim\mathcal{P}_\pi}\mathcal{L}(s')]] \geq 0$$
$$\mathbb{E}_{s\sim\mathcal{U}_\pi}[\mathcal{L}(s) - \lambda\mathbb{E}_{s'\sim\mathcal{P}_\pi}\mathcal{L}(s')] \leq 0 \qquad (24)$$

Since $\mathcal{L}(s) \geq c_\pi(s) + \lambda\mathbb{E}_{s'\sim\mathcal{P}_\pi}\mathcal{L}(s')$ holds, we get

$$\mathbb{E}_{s\sim\mathcal{U}_\pi}c_\pi(s) \leq 0 \qquad (25)$$

Based on the Abelian theorem, we know there exists

$$\mathcal{U}_\pi(s) = \lim_{T\to\infty} \frac{1}{T} \sum_{t=0}^{T} \mathcal{T}(s \mid \rho, \pi, t)$$
$$= \lim_{t\to\infty} \mathcal{T}(s|\rho,\pi,t) \qquad (26)$$
$$= \omega_\pi(s)$$

Thus, we get

$$\mathbb{E}_{s\sim\omega_\pi}[c_\pi(s)] \leq 0 \qquad (27)$$

The last row of inequality holds because of Equation (26). Based on the definition of $\omega_\pi(s)$, we have

$$\lim_{t\to\infty} \mathbb{E}_{\mathcal{T}(s|\rho,\pi,t)}c_\pi(s) \leq 0 \qquad (28)$$

Suppose that there exists a starting state $s_0 \in \{s_0 \mid c_\pi(s_0) \leq b\}$ and a positive constant $d$ such that $\lim_{t\to\infty} \mathbb{E}_{\mathcal{T}(s|\rho,\pi,t)}c_\pi(s) = d$ or $\lim_{t\to\infty} \mathbb{E}_{\mathcal{T}(s|\rho,\pi,t)}c_\pi(s) = \infty$. Consider that $\rho(s_0) > 0$ for all starting states in $\{s_0 \mid c_\pi(s_0) \leq b\}$ (Assumption C.1), then $\lim_{t\to\infty} \mathbb{E}_{s\sim\mathcal{T}(\cdot|\rho,\pi,t)}c_\pi(s) > 0$ , which is contradictory with Equation (28). Thus $\forall s_0 \in \{s_0 \mid c_\pi(s_0) \leq b\}, \lim_{t\to\infty} \mathbb{E}_{\mathcal{T}(s|\rho,\pi,t)}c_\pi(s) = 0$. Thus the system meets the mean cost stability by Definition 2.1.

### C.3 Comparison to Existing Methods

Intuitively, the previous method is a special case when $\mathcal{L}(s) = c_\pi(s) + \lambda\mathbb{E}_{s'\sim\mathcal{P}_\pi}\mathcal{L}(s')$ holds [9], as indicated in the following equation.

$$\mathbb{E}_{s\sim\mathcal{U}_\pi}[\mathbb{E}_{s'\sim\mathcal{P}_\pi}\mathcal{L}(s') - \mathcal{L}(s)] \leq \underbrace{-k[\mathbb{E}_{s\sim\mathcal{U}_\pi}[\mathcal{L}(s) - \lambda\mathbb{E}_{s'\sim\mathcal{P}_\pi}\mathcal{L}(s')]]}_{\text{Our method}}$$

$$\leq \underbrace{-k[c_\pi(s)]}_{\text{Han et al, 2020 [9]}} \qquad (29)$$

That means we extend the previous method to a more general case. To be specific, the introduction of $\lambda$ enlarges the solution space of the policy. Thus, it facilitates the policy to find the optimal point while maintaining the system's stability.

$\square$

## C.4 Finite-Time Feedback Tracking Method

**Lemma C.4** (Finite-Time Feedback Tracking Method). *In a continuous-time system, a trajectory $W(t)$ tracks the reference $R(t)$. $W(t)$ can track the reference within a finite time $T$, such that $R(t) = W(t), t \geq T$, if the following conditions holds.*

$$\nabla_t W(t) \leq -k(W(t) - R(t)), \forall t \in [0, T] \tag{30}$$

*Note that the gradient of $R(t)$ is bounded, meaning that $\nabla_t R(t) \leq \mu$ holds.*

*Proof.* First, we build the mean square error $V(t)$ between them.

$$V = \frac{1}{2}(W(t) - R(t))^2 \tag{31}$$

Then, we can derive the difference of $V(t)$ as follows

$$\begin{aligned}
\nabla_t V &= (W - R)(\nabla_t W - \nabla_t R) \\
&\leq (W - R)(-k(W - R) - \nabla_t R) \\
&\leq -k|W - R|^2 - (W - R)\nabla_t R
\end{aligned} \tag{32}$$

Introducing the Assumption that the bounded gradient of $R(t)$ , we have

$$\begin{aligned}
\nabla_t V &\leq -2k\frac{|W - R|^2}{2} - \sqrt{2}\mu\frac{|W - R|}{2} \\
&\leq -2kV - \sqrt{2}\mu\sqrt{V}
\end{aligned} \tag{33}$$

Observe that the above formulation belongs to a form of the Bernoulli differential equation. In this case, we can reduce the Bernoulli equation to a linear differential equation by substituting $z = \sqrt{V}$. Then, the general solution for $z$ is

$$z = \sqrt{V} \leq -\frac{\sqrt{2}}{2}\frac{\mu}{k} + Ce^{-kt} \tag{34}$$

Applying the initial condition $V(t = 0) = v_{t_0}$, we have

$$C = \sqrt{v_{t_0}} + \frac{\sqrt{2}}{2}\frac{\mu}{k} \tag{35}$$

Finally, the convergence time $T$ can be represented as:

$$T = \frac{1}{k}\ln\left(\frac{\frac{\sqrt{2}}{2}\frac{\mu}{k} + \sqrt{v_{t_0}}}{\frac{\sqrt{2}}{2}\frac{\mu}{k}}\right) + t_0 \tag{36}$$

$\square$

## C.5 Illustration of the Feedback Tracking

First, we denote $W(t)$ as $\mathcal{L}_\pi(s)$, $W(t + 1)$ as $\mathcal{L}_\pi(s')$ and $R(t)$ as $\lambda\mathcal{L}_\pi(s')$, where we omit the expection operator for simplicity. Specifically, at time $t + 1$, the value of $W(t + 1)$ should decrease by $k(W(t) - R(t))$. The change of $\lambda$ and $k$ results in $k(W(t) - R(t))$ increases correspondingly. Consequently, $W(t + 1)$ needs to decrease further to meet the requirement. Recalling the definition, $\mathcal{L}_\pi(s')$ become smaller. Additionally, the form of Equation (9) is similar to finite-time tracking method in continuous-time system which we depcit in Appendix C.4.

## C.6 Constrained Lyapunov Critic Network

Concretely, we denote the output of a neural network as $\mathbf{f}(s, a)$. And then, $\mathcal{L}_\theta(s, a)$ can be described by:

$$\mathcal{L}_\theta(s, a) = (\mathbf{G}_s(\mathbf{f}(s, a)))(\mathbf{G}_s(\mathbf{f}(s, a)))^\top \tag{37}$$

where $\mathbf{G}_s$ is a linear transformation, which guarantees $\mathbf{G}_{s=s_e}(\mathbf{f}(s = s_e, a)) = \mathbf{0}$ ($s_e$ is an equilibrium point defined in Definition B.1.). Note that $\mathbf{G}_s$ contains no parameters to be learned, so the operator does not cause harm to the representation ability of the neural network.

Concretely, the output of the neural network of the Lyapunov critic is described by:

$$\mathbf{f}(s, a) = \mathbf{h}_O(\mathbf{h}_{O-1}(\cdots \mathbf{h}_2(\mathbf{h}_1(< s, a >)))) \tag{38}$$

where each $h_o(z)$ has the same form:

$$\mathbf{h}_o(z) = \psi_o(\mathbf{W}_o z + \mathbf{b}_o) \tag{39}$$

Here, $O$ represents the number of layers, and $\psi_o$ is the non-linear activation function used in the $o$-th layer. Furthermore, $\{\mathbf{W}_o, \mathbf{b}_o\}$ is the weight and bias of the $o$-th layer.

First of all, to meet the demand of $\mathcal{L}_\theta(s_e, a) = 0$, we introduce a linear transformation $\mathbf{G}_s$, one of whose possible forms can be

$$\mathbf{G}_s(\mathbf{f}) = \frac{1}{\sum_i^I \delta s_i + \epsilon} \begin{bmatrix} \delta s_1 & \delta s_2 & \cdots & \delta s_I \end{bmatrix} \begin{bmatrix} \mathbf{f}_1 & \mathbf{f}_1 & \cdots & \mathbf{f}_v \\ \mathbf{f}_1 & \mathbf{f}_1 & \cdots & \mathbf{f}_v \\ \cdots & \cdots & \cdots & \cdots \\ \mathbf{f}_1 & \mathbf{f}_1 & \cdots & \mathbf{f}_v \end{bmatrix} \tag{40}$$

where $I$ denotes the number of elements of the state, and $v$ is the number of units of the output layer. $\epsilon$ is a constant close to 0 to avoid singularity. Note that $\delta s = s - s_e$, which indicates the difference between the current state and an equilibrium point. As we can see, when each element of $\delta s$ is zero, the multiplication of matrices is zero. Thus, $\mathbf{G}_{s=s_e}(\mathbf{f}(s = s_e, a)) = \mathbf{0}$ holds. Furthermore, it brings another benefit having no impact on the training of networks.

## C.7 Proof of Theorem 4.1

**Theorem C.5.** *Suppose that the length of sampling trajectories is $T$, then the bound can be expressed as:*

$$|\mathbb{E}_{s \sim \mathcal{U}_\pi} \Delta \mathcal{L}_\pi(s) - \mathbb{E}_{s \sim \mathcal{U}_\pi^T} \Delta \mathcal{L}_\pi(s)| \le 2 \frac{(k+1)\overline{c_\pi}}{1 - \gamma} T^{q-1} \tag{41}$$

*where $q$ is a constant in $(0, 1)$.*

*Proof.* First, we can get the following equation by introducing the definitions of $\mathcal{U}_\pi$ and $\mathcal{U}_\pi^T$.

$$\mathbb{E}_{s \sim \mathcal{U}_\pi} \Delta \mathcal{L}_\pi(s) - \mathbb{E}_{s \sim \mathcal{U}_\pi^T} \Delta \mathcal{L}_\pi(s)$$
$$= \int_{\mathcal{S}} (\mathcal{U}_\pi(s) - \frac{1}{T} \sum_{t=1}^T \mathcal{T}(s \mid \rho, \pi, t)) \Delta \mathcal{L}_\pi(s) \mathrm{d}s \tag{42}$$
$$= \frac{1}{T} \sum_{t=1}^T \int_{\mathcal{S}} (\mathcal{U}_\pi(s) - \mathcal{T}(s \mid \rho, \pi, t)) \Delta \mathcal{L}_\pi(s) \mathrm{d}s$$

Then, eliminating the integral operator, we obtain

$$|\mathbb{E}_{s \sim \mathcal{U}_\pi} \Delta \mathcal{L}_\pi(s) - \mathbb{E}_{s \sim \mathcal{U}_\pi^T} \Delta \mathcal{L}_\pi(s)|$$
$$\le \frac{1}{T} \sum_{t=1}^T \|\mathcal{U}_\pi(s) - \mathcal{T}(s \mid \rho, \pi, t)\|_1 \|\Delta \mathcal{L}_\pi(s)\|_\infty \tag{43}$$

Thus, the next step is to get the bounds of $\|\mathcal{U}_\pi(s) - \mathcal{T}(s \mid \rho, \pi, t)\|_1$ and $\|\Delta \mathcal{L}_\pi(s)\|_\infty$.

For the first part, we introduce the assumption that first is mentioned in [38], shown as follows:

$$\sum_{t=1}^{T} \|\mathcal{U}_\pi(s) - \mathcal{T}(s \mid \rho, \pi, t)\|_1 \leq 2T^q, \quad \forall T \in \mathcal{Z}_+, \ \exists q \in (0, 1) \tag{44}$$

Frankly speaking, the assumption is easily satisfied because the L1 distance between two distributions is bounded by 2. At the same time, $\mathcal{T}(s \mid \rho, \pi, t)$ converges to $\mathcal{U}_\pi(s)$ with time approaching.

For the second part, we can get the bound of $\Delta\mathcal{L}_\pi(s)$ according to Equation 17.

$$\begin{aligned}
\Delta\mathcal{L}_\pi(s) &= \mathbb{E}_{s' \sim \mathcal{P}_\pi}\mathcal{L}_\pi(s') - \mathcal{L}_\pi(s) + k(\mathcal{L}_\pi(s) - \lambda\mathbb{E}_{s' \sim \mathcal{P}_\pi}(s')) \\
&\leq \frac{\overline{c_\pi}}{1 - \gamma} - 0 + k(\frac{\overline{c_\pi}}{1 - \gamma} - 0)
\end{aligned} \tag{45}$$

Then, we have

$$\|\Delta\mathcal{L}_\pi(s)\|_\infty \leq (k + 1)\frac{\overline{c_\pi}}{1 - \gamma} \tag{46}$$

Adding results in Equation 47, we finally get

$$|\mathbb{E}_{s \sim \mathcal{U}_\pi}\Delta\mathcal{L}_\pi(s) - \mathbb{E}_{s \sim \mathcal{U}_\pi^T}\Delta\mathcal{L}_\pi(s)| \leq 2\frac{(k + 1)\overline{c_\pi}}{1 - \gamma}T^{q-1} \tag{47}$$

$\square$

### C.8 Proof of Theorem 4.2

**Theorem C.6.** *Suppose that the length of sampling trajectories is $T$ and the number of trajectories is $M$, then there exists the following upper bound:*

$$\begin{aligned}
\mathbb{P}(|\frac{1}{MT}\sum_{m=1}^{M}\sum_{t=1}^{T}\Delta\mathcal{L}_\pi(s_t^m) - \mathbb{E}_{s \sim \mathcal{U}_\pi^T}\Delta\mathcal{L}_\pi(s)| \geq \alpha) \\
\leq 2\exp(-\frac{M\alpha^2(1 - \gamma)^2}{((1 - k\lambda)^2 + (k - 1)^2)\overline{c_\pi}^2})
\end{aligned} \tag{48}$$

*where $s_t^m$ represents the state in the $m$-th trajectory at the timestep $t$.*

*Proof.* First, eliminating $\Delta\mathcal{L}_\pi(s)$ by Equation 9, we rewrites the left side of Equation 48 as

$$\begin{aligned}
\delta &= \mathbb{P}(|\frac{1}{MT}\sum_{m=1}^{M}\sum_{t=1}^{T}\Delta\mathcal{L}_\pi(s_t^m) - \mathbb{E}_{s \sim \mathcal{U}_\pi^T}\Delta\mathcal{L}_\pi(s)| \geq \alpha) \\
&= \mathbb{P}(|\frac{1}{MT}\sum_{m=1}^{M}\sum_{t=1}^{T}(\mathcal{L}_\pi(s_{t+1}) - \mathcal{L}_\pi(s_t) + k_l(\mathcal{L}_\pi(s_t) - \lambda\mathcal{L}_\pi(s_{t+1}))) - \mathbb{E}_{s \sim \mathcal{U}_\pi^T}\Delta\mathcal{L}_\pi(s)| \geq \alpha) \\
&= \mathbb{P}(|\frac{1}{MT}\sum_{m=1}^{M}\sum_{t=1}^{T}((1 - k\lambda)\mathcal{L}_\pi(s_{t+1}) + (k - 1)\mathcal{L}_\pi(s_t)) - \mathbb{E}_{s \sim \mathcal{U}_\pi^T}\Delta\mathcal{L}_\pi(s)| \geq \alpha)
\end{aligned} \tag{49}$$

Here $\mathbb{E}_{s \sim \mathcal{U}_\pi^T}\Delta\mathcal{L}_\pi(s)$ is expected value of $\frac{1}{MT}\sum_{m=1}^{M}\sum_{t=1}^{T}\Delta\mathcal{L}_\pi(s_t^m)$. In addition, the bounds of $(1 - k\lambda)\mathcal{L}_\pi(s_{t+1})$ and $(k - 1)\mathcal{L}_\pi(s_t)$ can be obtained easily by Equation 17. Thus, we obtain the Theorem 4.2 by applying Hoeffding's inequality.

$$\begin{aligned}
\delta &\leq 2\exp(-\frac{2M^2\alpha^2}{M((1 - k\lambda)^2 + (k - 1)^2)\frac{\overline{c_\pi}^2}{(1-\gamma)^2}}) \\
&\leq 2\exp(-\frac{M\alpha^2(1 - \gamma)^2}{((1 - k\lambda)^2 + (k - 1)^2)\overline{c_\pi}^2})
\end{aligned} \tag{50}$$

$\square$

# D    Details of Algorithms

As mentioned in the main text, we introduce a minimum entropy as a constraint in policy optimization and apply the primal-dual method to update the policy and the Lagrange multiplier $\lambda_e$. To be specific, the constraint can be expressed as

$$\log \pi_\phi(a|s) \leq -\mathcal{Z}_e \tag{51}$$

where $\mathcal{Z}_e$ is the minimum value of policy entropy, usually, $\mathcal{Z}_e$ corresponds to the dimension of action space in the environment.

---

**Algorithm 1:** Adaptive Lyapunov-based Actor-Critic Algorithm (ALAC)

---

    Orthogonal initialize the parameters of actor and critic networks with $\phi, \theta$
    Initialize replay buffer $D$ and $\lambda_l, \lambda_e, \lambda$ and $k$
    Initialize the parameters of target network with $\phi' \leftarrow \phi$ and $\theta' \leftarrow \theta$
    **for** episode $m = 1, M$ **do**
      Sample an initial state $s_0$
      **for** step $t = 0, T - 1$ **do**
        Sample an action from $\pi_\phi(a_t|s_t)$
        Execute the action $a_t$ and observe a new state $s_{t+1}$
        Store $< s_t, a_t, c_t, s_{t+1} >$ into $\mathcal{D}$
      **end for**
      **for** iteration $n = 1, N$ **do**
        Sample a minibatch $\mathcal{B}$ from the replay buffer $D$
        Update $\theta$ according to Eq.(10) using minibatch $\mathcal{B}$
        Update $\phi, \lambda_l, \lambda_e$ according to Eq.(11),(12),(51) using minibatch $\mathcal{B}$
        Update adaptive factors $\lambda$ and $k$
        Update the parameters of target networks, $\theta', \phi'$.
      **end for**
    **end for**

---

# E    Details of Experiments

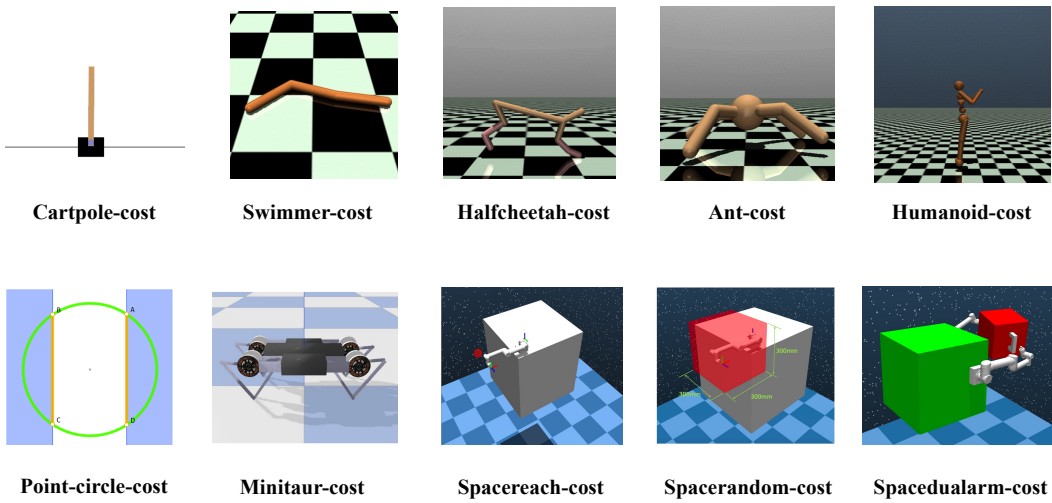

Figure 5: Overview of our environments.

We test our method and baselines in ten robotic control environments, including **Cartpole-cost**, **Point-circle-cost**, **Halfcheetah-cost**, **Swimmer-cost**, **Ant-cost**, **Humanoid-cost**, **Minitaur-cost**, **Spacereach-cost**, **Spacerandom-cost** and **Spacedualarm-cost**. Most tasks in ten environments are goal-oriented, tracking a target position or speed, which corresponds to most control tasks. Furthermore, the latter four environments involve models of practical robots like a quadruped robot and a

robotic arm, making them relatively more difficult. It is worth noting that the task of **Spacedualarm-cost** is trajectory planning of a free-floating dual-arm space robot. The coupling property of the base and the robotic arms brings hardship for both traditional control and RL-based methods [39].

## E.1 Environmental Design

**Cartpole-cost** This task aims to maintain the pole vertically at a target position. The environment is inherited from [33]. The state and action space are the same as the default settings in OpenAI Gym[40], so we omit the description. The cost function is $c = \left(\frac{x}{x_{\text{threshold}}}\right)^2 + 20 * \left(\frac{\theta}{\theta_{\text{threshold}}}\right)^2$, where $x_{\text{threshold}} = 10$ and $\theta_{\text{threshold}} = 20°$. The other settings can be found in Table 3.

**Point-circle-cost** This task aims to allow a sphere to track a circular trajectory. The environment is inherited from [41]. The sphere is initialized at the original point. The cost function is represented as $c = d$, where $d$ denotes the distance between the current position and the reference. The other settings can be found in Table 3.

Table 3: Hyper-parameters of non-linear dynamic environments

| Hyper-parameters | Cartpole-cost | Point-circle-cost |
|---|---|---|
| State shape | 4 | 7 |
| Action shape | 2 | 2 |
| Length of an episode | 250 steps | 65 steps |
| Maximum steps | 3e5 steps | 3e5 steps |
| Actor network | (64, 64) | (64, 64) |
| Critic network | (64, 64, 16) | (64, 64, 16) |

**Halfcheetah-cost** The goal of this task is to make a HalfCheetah (a 2-legged simulated robot) to track the desired velocity. The environment is inherited from [33]. The state and action space are the same as the default settings in OpenAI Gym[40], so we omit the description. The cost function is $c = (v - 1)^2$, where 1 represents the desired velocity. The other settings can be found in Table 4.

**Swimmer-cost** This task aims to make a multi-joint snake robot to track the desired velocity. The environment is inherited from [33]. The state and action space are the same as the default settings in OpenAI Gym[40], so we omit the description. The cost function is $c = (v - 1)^2$, where 1 represents the desired velocity. The other settings can be found in Table 4.

**Ant-cost** This task aims to make an Ant (a quadrupedal simulated robot) track the desired velocity. The environment is inherited from [40]. The state and action space are the same as the default settings in OpenAI Gym [40], so we omit the description. The cost function is $c = (v - 1)^2$, where 1 represents the desired velocity. The other settings can be found in Table 4.

**Humanoid-cost** This task aims to make a humanoid robot to track the desired velocity. The environment is inherited from [40]. The state and action space are the same as the default settings in OpenAI Gym [40], so we omit the description. The cost function is $c = (v - 1)^2$, where 1 represents the desired velocity. The other settings can be found in Table 4.

**Minitaur-cost** This task aims to control the Ghost Robotics Minitaur quadruped to run forward at the desired velocity. The environment is inherited from [42]. The state and action space are the same as the default settings in PyBullet environment[42], so we omit the description. The cost function is $c = (v - 1)^2$, where 1 represents the desired velocity. The other settings can be found in Table 5.

**Spacereach-cost** This task aims to make a free-floating single-arm space robot's end-effector reach a fixed goal position. Since the base satellite is uncontrolled, collisions will cause system instability once collisions occur. Therefore, it is critical to plan a collision-free path while maintaining the

Table 4: Hyper-parameters of mujoco environments

| Hyper-parameters | Swimmer-cost | Halfcheetah-cost | Ant-cost | Humanoid-cost |
|---|---|---|---|---|
| State shape | 8 | 17 | 27 | 376 |
| Action shape | 2 | 6 | 8 | 8 |
| Length of an episode | 250 steps | 200 steps | 200 steps | 500 steps |
| Maximum steps | 3e5 steps | 1e6 steps | 1e6 steps | 1e6 steps |
| Actor network | (64, 64) | (64, 64) | (64, 64) | (256, 256) |
| Critic network | (64, 64, 16) | (256, 256, 16) | (64, 64, 16) | (256, 256, 128) |

Table 5: Hyper-parameters of robotic environments

| Hyper-parameters | Minitaur-cost | Spacereach-cost | Spacerandom-cost | Spacedualarm-cost |
|---|---|---|---|---|
| State shape | 27 | 18 | 18 | 54 |
| Action shape | 8 | 6 | 6 | 12 |
| Length of an episode | 500 steps | 200 steps | 200 steps | 200 steps |
| Maximum steps | 1e6 steps | 3e5 steps | 5e5 steps | 5e5 steps |
| Actor network | (256, 256) | (256, 256) | (256, 256) | (512, 512) |
| Critic network | (256, 256, 16) | (256, 256, 128) | (256, 256, 128) | (512, 512, 256) |

stability of the base. The agent can obtain the state, including the angular positions and velocities of joints, the position of the end-effector, and the position of the reference point. Then, the agent outputs the desired velocities of joints. In low-level planning, a PD controller converts the desired velocities into torques, and then controls the manipulator. The cost function is defined as $c = d$, where $d$ is the distance between the goal and end-effector. The other settings can be found in Table 5.

**Spacerandom-cost**    This task aims to make a free-floating single-arm space robot's end-effector reach a random goal position. The agent can obtain the state, including the angular positions and velocities of joints, the position of the end-effector, and the position of the reference point. Then, the agent outputs the desired velocities of joints. In low-level planning, a PD controller converts the desired velocities into torques to control the manipulator. The cost function is defined as $c = d$, where $d$ is the distance between goal and end-effector. The other settings can be found in Table 5.

**Spacedualarm-cost**    This task aims to make a free-floating dual-arm space robot's end-effectors reach random goal positions. The complexity of the task increases dramatically due to two arms' coupling effects on the base. The agent can obtain the state, including the angular positions and velocities of joints, the positions of end-effectors, and the position of target points of two manipulators. Then, the agent outputs the desired velocities of joints. In low-level planning, a PD controller converts the desired velocities into torques to control the manipulators. The cost function is defined as follows: $c = d_0 + d_1$, where $d_i$ is the distance between goal and end-effector of Arm-$i$. The other settings can be found in Table 5.

### E.2    Implementation Details

#### E.2.1    Baselines

**SAC-cost**    Soft Actor-Critic (SAC) is an off-policy maximum entropy actor-critic algorithm [19]. The main contribution is to add a maximum entropy objective into standard algorithms. The soft Q and V functions are trained to minimize the soft Bellman residual, and the policy can be learned by directly minimizing the expected KL-divergence. The only difference between SAC and SAC-cost is replacing maximizing a reward function with minimizing a cost function. The hyper-parameters of **SAC-cost** is illustrated in Table 6.

**SPPO**    Safe proximal policy optimization (SPPO) is a Lyapunov-based safe policy optimization algorithm. The neural Lyapunov network is constructed to prevent unsafe behaviors. Actually, the

Table 6: Hyper-parameters of **SAC-cost**

| Hyper-parameters | SAC-cost |
|---|---|
| Learning rate of actor | 1.e-4 |
| Learning rate of critic | 3.e-4 |
| Optimizer | Adam |
| ReplayBuffer size | $10^6$ |
| Discount ($\gamma$) | 0.995 |
| Polyak ($1 - \tau$) | 0.995 |
| Entropy coefficient | 1 |
| Batch size | 256 |

safe projection method is inspired by the TRPO algorithm [43]. In this paper, we modify it to apply the Lyapunov constraints on the MDP tasks, similar to the process in [9]. The hyper-parameters of **SPPO** is illustrated in Table 7.

Table 7: Hyper-parameters of **SPPO**

| Hyper-parameters | SPPO |
|---|---|
| Learning rate of actor | 1.e-4 |
| Learning rate of Lyapunov | 3.e-4 |
| Optimizer | Adam |
| Discount ($\gamma$) | 0.995 |
| GAE parameter ($\lambda$) | 0.95 |
| Clipping range | 0.2 |
| KL constraint ($\delta$) | 0.2 |
| Fisher estimation fraction | 0.1 |
| Conjugate gradient steps | 10 |
| Conjugate gradient damping | 0.1 |
| Backtracking steps | 10 |
| Timesteps per iteration | 2000 |

**LAC**    Lyapunov-based Actor-Critic(LAC) algorithm is an actor-critic RL-based algorithm jointly learning a neural controller and Lyapunov function [9]. Particularly, they propose a data-driven stability condition on the expected value over the state space. Moreover, they have found that the method achieves high generalization and robustness. The hyper-parameters of **LAC** is illustrated in Table 8. Among them, $\alpha_3$ is 0.1 in **LAC**, while it is changed as 1 in **LAC**$^*$.

Table 8: Hyperparameters of **LAC**

| Hyperparameters | LAC |
|---|---|
| Learning rate of actor | 1.e-4 |
| Learning rate of Lyapunov | 3.e-4 |
| Learning rate of Larange multiplier | 3.e-4 |
| Optimizer | Adam |
| ReplayBuffer size | $10^6$ |
| Discount ($\gamma$) | 0.995 |
| Polyak ($1 - \tau$) | 0.995 |
| Parameter of Lyapunov constraint ($\alpha_3$) | 0.1 |
| Batch size | 256 |

**POLYC**    Policy Optimization with Self-Learned Almost Lyapunov Critics (POLYC) algorithm is built on the standard PPO algorithm [44]. Introducing a Lyapunov function without access to the cost allows the agent to self-learn the Lyapunov critic function by minimizing the Lyapunov risk. The hyper-parameters of **POLYC** is illustrated in Table 9.

Table 9: Hyper-parameters of **POLYC**

| Hyper-parameters | POLYC |
|---|---|
| Learning rate of actor | 1.e-4 |
| Learning rate of critic | 3.e-4 |
| Learning rate of Lyapunov | 3.e-4 |
| Optimizer | Adam |
| Discount ($\gamma$) | 0.995 |
| GAE parameter ($\lambda$) | 0.95 |
| Weight of Lyapunov constraint ($\beta$) | 0.1 |
| Clipping range | 0.2 |
| Timesteps per iteration | 2000 |

**LBPO**  Lyapunov Barrier Policy Optimization (LBPO) algorithm [17] is built on SPPO algorithm [18]. However, the core improvement uses a Lyapunov-based barrier function to restrict the policy update to a safe set for each training iteration. Compared with the SPPO algorithm, the method avoids backtracking to ensure safety. For the implementation in our paper, the process is similar to that of the SPPO algorithm. The hyperparameters of **LBPO** is illustrated in Table 10.

Table 10: Hyperparameters of **LBPO**

| Hyperparameters | LBPO |
|---|---|
| Learning rate of actor | 1.e-4 |
| Learning rate of critic | 1.e-4 |
| Learning rate of Lyapunov | 3.e-4 |
| Optimizer | Adam |
| Discount ($\gamma$) | 0.99 |
| GAE parameter ($\lambda$) | 0.97 |
| Clipping range | 0.2 |
| KL constraint | 0.012 |
| Fisher estimation fraction | 0.1 |
| Conjugate gradient steps | 10 |
| Conjugate gradient damping | 0.1 |
| Backtracking steps | 10 |
| Weight of Lyapunov constraint ($\beta$) | 0.01 |
| Timesteps per iteration | 2000 |

**TNLF**  Twin Neural Lyapunov Function (TNLF) algorithm is proposed to deal with safe robot navigation in [11]. Different from other approaches, the TNLF method defines a Lyapunov V function and Lyapunov Q function, which are trained by minimizing the Lyapunov risk. In effect, the Lyapunov risk is similar to that of [10]. Since the Lyapunov function strictly decreases over time, the robot starting with any state in a Region of Attraction (RoA) will always stay in the RoA in the future. It should be pointed out that as our environments only support the cost function, the objective, except for Lyapunov risk, is to minimize the cumulative return of cost. The hyper-parameters of **TNLF** is illustrated in Table 11.

### E.2.2 Our method

**ALAC**  Our method offers a significant advantage in contrast to baselines, which is to use fewer hyperparameters. The main hyperparameters are illustrated in Table 12. We notice that these parameters control networks' learning without including the parameters of constraints. The reason is they are automatically updated according to Lagrange multipliers, $\lambda_l$, and $\lambda_e$. The initial value of Lagrange multipliers is set to 1, common usage in previous constrained methods.

Table 11: Hyper-parameters of **TNLF**

| Hyper-parameters | TNLF |
|---|---|
| Learning rate of actor | 1.e-4 |
| Learning rate of critic | 3.e-4 |
| Learning rate of Lyapunov V functiob | 3.e-4 |
| Learning rate of Lyapunov functiob | 3.e-4 |
| Optimizer | Adam |
| ReplayBuffer size | $10^6$ |
| Discount ($\gamma$) | 0.995 |
| Polyak ($1 - \tau$) | 0.995 |
| Weight of Lyapunov constraint ($\alpha$) | 0.1 |
| Variance of noise distribution | 1 |
| Batch size | 256 |

Table 12: Hyper-parameters of **ALAC**

| Hyper-parameters | ALAC |
|---|---|
| Learning rate of actor | 1.e-4 |
| Learning rate of Lyapunov | 3.e-4 |
| Learning rate of Lagrange multipliers ($\lambda_l$ and $\lambda_e$) | 3.e-4 |
| Optimizer | Adam |
| ReplayBuffer size | $10^6$ |
| Discount ($\gamma$) | 0.995 |
| Polyak ($1 - \tau$) | 0.995 |
| Batch size | 256 |

### E.3    More Results on Comparison

Figure 10 shows the learning curves of the accumulated cost and constraint violations of **ALAC** and other baselines in ten environments.

Note that Eq. (3) illustrates the definition of optimal-time stability. We can observe that when the system satisfies the stability condition and the accumulated cost is minimal, the system achieves optimal-time stability. Therefore, in our experiments, we use the accumulated cost in a testing episode as the metric of optimality and the stability constraint violations as the stability metric. As shown in Table 1, our method achieves both minimal accumulated costs and minimal stability violations compared with the baselines.

**violation of stability conditions.** We measure the stability of each algorithm by evaluating the violations of stability conditions in a trajectory (episode). Since different algorithms may have different requirements for Lyapunov functions, we cannot use a single metric function as the stability metric. However, all algorithms aim to minimize the stability constraints. When the value of these constraints is not close to 0, it indicates that the algorithm does not satisfy the designed stability conditions. In such cases, the algorithm cannot guarantee the convergence of the system's state to an equilibrium point.

### E.4    More Results on Ablation Study

We provide the specific formulation of $\Delta\mathcal{L}^1_{\pi_\phi}$ and $\Delta\mathcal{L}^2_{\pi_\phi}$. Compared with $\Delta\mathcal{L}_{\pi_\phi}$ in Equation 9, we intuitively find that $\Delta\mathcal{L}^1_{\pi_\phi}$ and $\Delta\mathcal{L}^2_{\pi_\phi}$ are lower and higher bound of $\Delta\mathcal{L}_{\pi_\phi}$ respectively. In other words, $\Delta\mathcal{L}^1_{\pi_\phi}$ represents the strongest constraint, while $\Delta\mathcal{L}^2_{\pi_\phi}$ represents the loosest constraint. The comparison between them can demonstrate that the sampling-based Lyapunov stability ($\Delta\mathcal{L}_{\pi_\phi}$) can search for the optimal policy with stability guarantee due to the adaptive updating of $\lambda$.

```
1   n_components=2 or 3,
2   early_exaggeration=12,
3   learning_rate=200.0,
4   n_iter=1000,
5   n_iter_without_progress=300,
6   min_grad_norm=1e-7,
7   perplexity=30,
8   metric="euclidean",
9   n_jobs=None,
10  random_state=42,
11  verbose=True,
12  init='pca'
```

Table 13: Other hyper-parameters of t-SNE method.

$$\Delta\mathcal{L}^1_{\pi_\phi}(s,a) = \mathcal{L}_\theta(s', \pi_\phi(\cdot|s')) - \mathcal{L}_\theta(s,a) + k[\mathcal{L}_\theta(s,a) - 0]$$
$$\Delta\mathcal{L}^2_{\pi_\phi}(s,a) = \mathcal{L}_\theta(s', \pi_\phi(\cdot|s')) - \mathcal{L}_\theta(s,a) + k[\mathcal{L}_\theta(s,a) - \mathcal{L}_\theta(s', \pi_\phi(\cdot|s'))] \qquad (52)$$
$$\Delta\mathcal{L}_{\pi_\phi}(s,a) = \mathcal{L}_\theta(s', \pi_\phi(\cdot \mid s')) - \mathcal{L}_\theta(s,a) + k[\mathcal{L}_\theta(s,a) - \lambda\mathcal{L}_\theta(s', \pi_\phi(\cdot \mid s'))]$$

Furthermore, we also conduct the algorithm with a constant $k = 0.1$. The comparison demonstrates that the heuristic-based updating of $k$ is an effective method. The ablation experiments on other tasks are shown in Figure 9.

### E.5 Details of Visualization

Our RL-based policy optimization method guided by adaptive stability is difficult to express the latent laws of states in the convergent process of different environments as the high-dimension states-space. To find and show the state's change laws in the convergent process:

- We use the t-SNE dimension reduction technique to visualize the state-space.

- We plot the phase trajectory with variance according to the state pairs of joint angular position and velocity.

- We plot the Lyapunov-value surface and its shadow with the phase trajectory and values in the convergence process.

**T-SNE Visualization** The top row of Figure 3 shows the results of the t-SNE state plotting with SciKit-Learn tools(i.e.sklearn.manifold.TSNE function) with varying parameters(e.g. early_exaggeration, min_ grad_norm). Cartpole-Cost is visualized with n_components=2 while other environments with n_components=3. The hyper-parameters for t-SNE are shown in Table 13.

**Phase Trajectories of Systems** We select the angular position and velocity of a joint in the state space in each environment and plot the phase trajectory with variance in Figure 6. The convergent process is shown as the angular velocity starts from 0 to 0, and the joint angle starts from the beginning to the convergence position.

**Lyapunov Functions of Systems** We visualize the change of Lyapunov-value in 3 dimensions based on the phase trajectory. The second row of Figure 3 shows the Lyapunov-value surface. The curves of values along the phase trajectory are mapped to the whole plane with down-sampled and smoothed by a Gaussian filter; we add the values and the phase trajectory shadows correspondingly simultaneously.

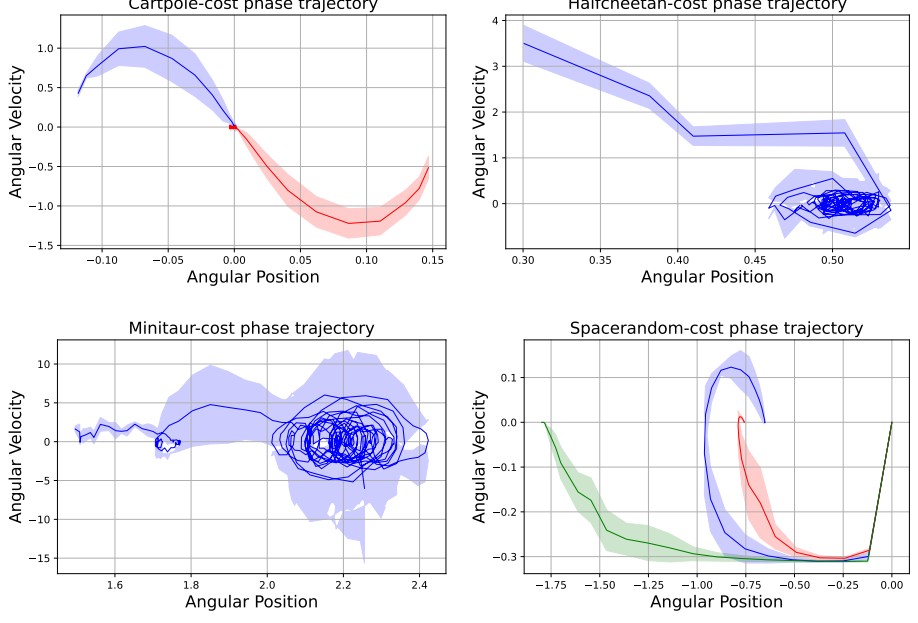

Figure 6: Phase trajectories of the systems trained by ALAC. (we report the results of 20 trials and select a joint to graph the phase trajectory in each task.)

## E.6 More Results on Evaluation

### E.6.1 Robustness

Generally speaking, stability has a potential relationship with robustness to some extent. Thus, we add external disturbances with different magnitudes in each environment and observe the performance difference. Specifically, when evaluating, we introduce an external disturbance on actions every 50 steps. The magnitudes determine the range of disturbances. For example, a magnitude equal to 0.5 means that the noise is randomly picked from [-0.5, 0.5]. Figure 8 shows that in all scenarios, **ALAC** outperforms other methods under different disturbance magnitudes. Furthermore, we omit the algorithms that do not converge to a reasonable solution in each task.

### E.6.2 Generalization

We verify that **ALAC** achieves excellent generalization with the feedback of the error. First, we introduce a new variable in the state, which represents the error between the desired goal and the achieved goal. To be specific, the desired goal can be the desired velocity or position of the agent, and the achieved goal is the current velocity or position of the agent. During training, we use a fixed value for the desired goal, while in evaluation, we add $\pm 20\%$ bias to the desired goal. The results show that **ALAC** generalizes well to previously unseen desired goals, as depicted in Figure 8. Furthermore, we observe that errors have a negative impact on the performance of **SAC-cost**. The reason could be that **SAC-cost** does not capture the error information without the guidance of a Lyapunov function. Notably, the number of environment steps in **Halfcheetah-cost** is 5e5 in this section.

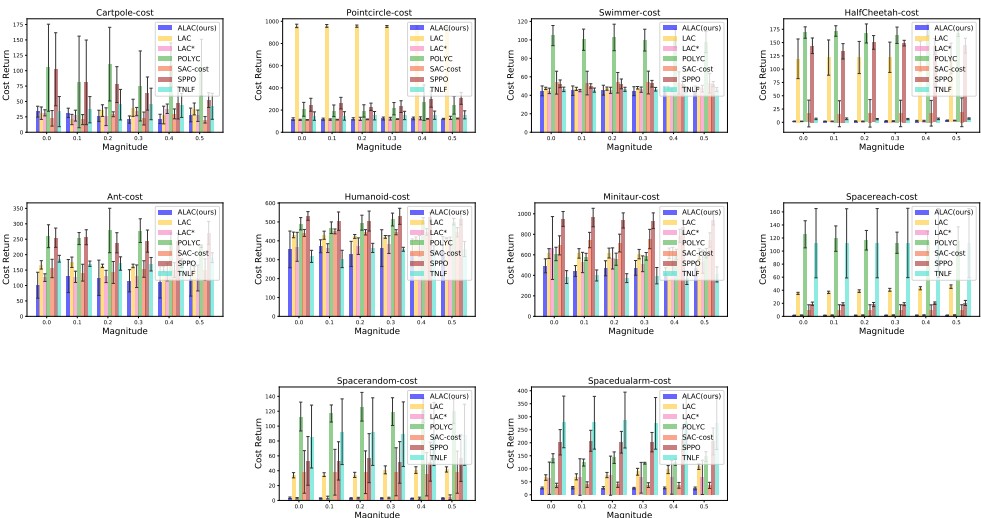

Figure 7: Performance of ALAC method and other baselines under persistent disturbances with different magnitudes. (The X-axis indicates the magnitude of the applied disturbance. We evaluate the trained policies for 20 trials in each setting.)

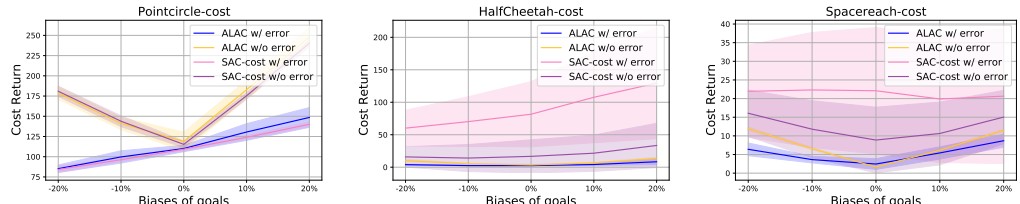

Figure 8: Evaluation of ALAC and SAC-cost methods in the presence of different biases of goals. (The X-axis indicates the magnitude of the applied shifting. We evaluate the trained policies for 20 trials in each setting.)

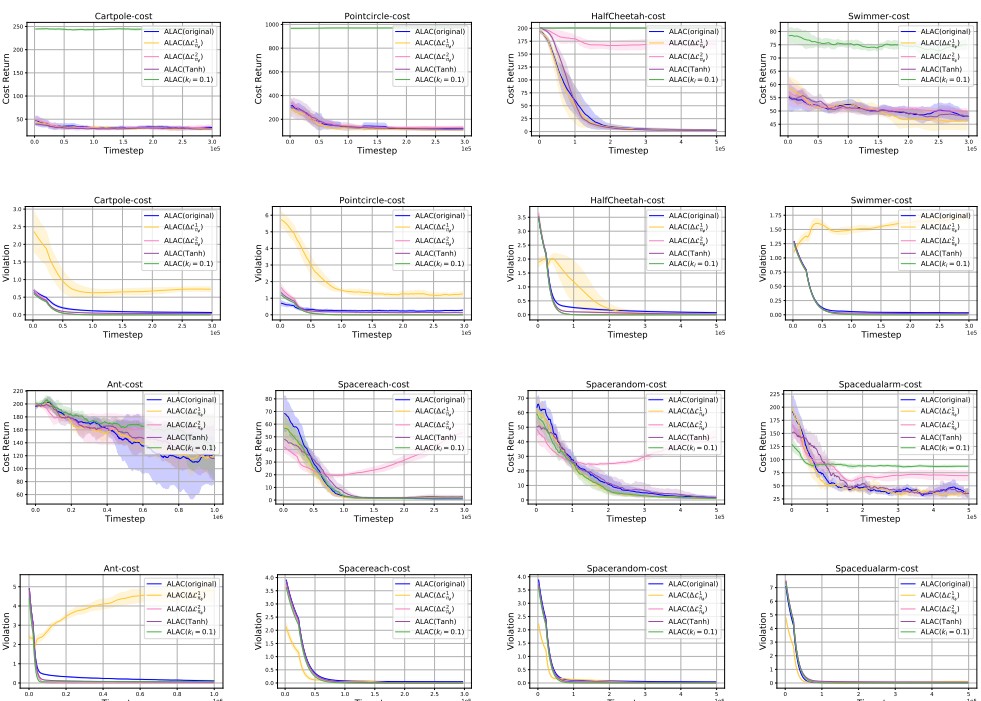

Figure 9: Ablation studies of our method. ALAC(original) shows comparable or the best performance compared with other certifications on each task.

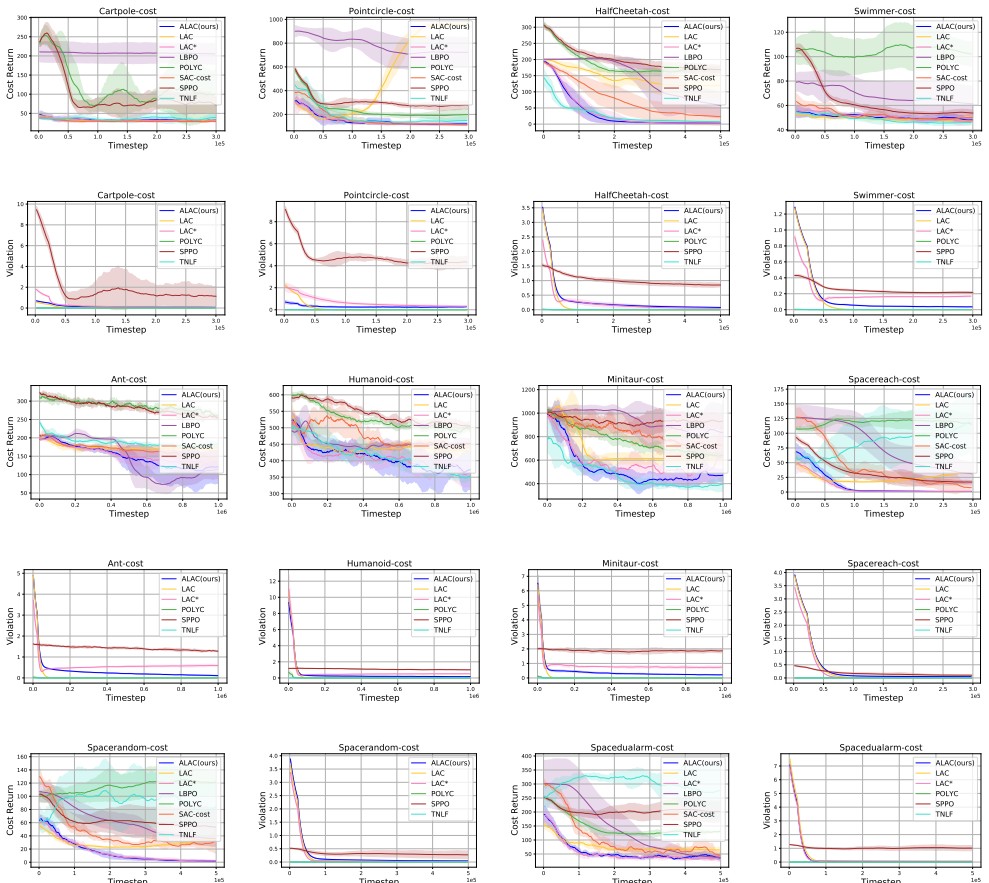

Figure 10: Performance comparison on ten tasks. The ALAC method finds a good trade-off between minimizing the accumulated cost and constraint violations in contrast to their rivals.

