# OpenReview forum: "A Policy Optimization Method Towards Optimal-time Stability"
_robot-learning.org/CoRL/2023/Conference — CoRL 2023 Poster_

### Official Review · Reviewer_bi85 · 2023-07-19

**Confidence:** 3
**Originality:** Good
**Technical Quality:** Very Good
**Clarity Of Presentation:** Good
**Impact:** 3

**Recommendation:**

Weak Accept: I recommend accepting the paper, but will not argue for my recommendation if the majority of other reviewers have a different opinion.

**Review:**

The paper provides strong and detailed theoretical results on optimization of accumulated cost under stability constraints.
The paper demonstrates the superiority of the proposed algorithm compared to existing baselines on a series of RL robotics tasks in simulated environments.

It would be extremely valuable if the author could clearly state what are the approximations that are made when transitioning from the theory to the actual algorithm. What are the choices dictated by the theoretical analysis and which choices are arbitrary, or heuristics that tend to work well in practice.

For instance, why is the Lagrange multiplier lambda clipped between 0 and 1? Enforcing positive multiplier is expected for inequality constraints, however it is unclear why the algorithm requires clipping multiplier values larger than 1. The choice seems somewhat arbitrary. It would be great if the authors could motivate this choice with additional details.

Overall, it is hard for the reader to distinguish between the algorithm design choices that are directly stemming from the theoretical analysis and the ones that are only loosely related to the theory section or simple heuristics.

Minor issues:
cultivated costs  <- accumulated costs
satisfy focal stability <- local stability

**Quality Of The Limitations Section:**

Additional details required

**Questions For Rebuttal:**

To improve the clarity of the paper, it would be great if the authors could explain which algorithm design choices are directly stemming from the theoretical analysis and the ones that are only loosely related to the theory section or simple heuristics.

**Robotics Focus:**

Relevant but unlikely to deploy to hardware in near future

**Summary Of Paper:**

In this paper, the authors propose a novel model-free reinforcement learning (RL) algorithm. This algorithm (ALAC) borrows techniques from Lyapunov stability analysis to achieve optimal behavior under some stability constraints. The proposed approach is integrated within the Actor-Critic framework. The method is motivated by a theoretical analysis, the algorithm implementation is derived from the theoretical analysis. Finally, the algorithm is successfully implemented on 10 RL tasks related to robotics problems. Comparisons with existing RL algorithms baselines are provided.

**Summary Of Recommendation:**

This paper propose a new algorithm to solve RL problems under stability constraints. The proposed approach is backed by theoretical results and is benchmarked under a number of baselines on a set of robotics tasks in simulation.

---

### Official Review · Reviewer_fiUo · 2023-07-20

**Confidence:** 3
**Originality:** Very Good
**Technical Quality:** Very Good
**Clarity Of Presentation:** Very Good
**Impact:** 4

**Recommendation:**

Weak Accept: I recommend accepting the paper, but will not argue for my recommendation if the majority of other reviewers have a different opinion.

**Review:**

The paper presents a generalized approach to sThe paper presents a generalized approach to incorporating convergence stability in policy optimization. The lyapunov critic follows standard Q-function formulation but with cost (instead of reward) and the goal is to minimize the state-action value. The Lagrangian constraint-based policy update rule is intuitive and theoretically justified.

The algorithm is compared with relevant baselines using Lyapunov based formulations. The analysis on stability constraint violations is not explained. Is the same metric used for all the algorithms? What is the metric?

The paper mentions optimal time convergence, how is the convergence time optimal? Does that mean ALAC is sample efficient as compared to the baselines? Such a conclusion cannot be drawn from Fig 10. Can you clarify?

For generalization experiments, how do you calculate the errors like what is a desired goal and an achieved goal? What is a reference signal? Can you please clarify further, maybe, with an example from an environment?

**Quality Of The Limitations Section:**

Limitations are addressed clearly

**Questions For Rebuttal:**

See review above.

**Robotics Focus:**

Relevant but unlikely to deploy to hardware in near future

**Summary Of Paper:**

The paper presents a Lyapunov-based approach for policy optimization. The state-action value function (critic) is replaced by a Lyapunov candidate, the expected discounted sum of costs, under certain MDP stability conditions. The critic can then be trained by optimizing the standard TD-learning objective. The policy update is formulated to achieve cost optimality such that the cost incurred at a step tends to zero while satisfying stability conditions added with suitable Lagrange multipliers. The multipliers are updated to increase the strength of the constraints gradually and reach the equilibrium state. A constrained critic network is used such that the Lyapunov value of state action is always positive, zero if the state is an equilibrium point. The authors compare the proposed setup with prior lyapunov-based methods on ten simulated environments. Ablations are shown to study the importance of the strength of the stability constraint.

**Summary Of Recommendation:**

I believe that the paper introduces an interesting direction in using lyapunov-based formulations for policy optimization in RL. It introduces new stability conditions and establishes maximum deviation bounds w.r.t. practical implementation. The comparison results are also extensive. However, the results are not convincing from a robotics perspective, as for both Minitaur and Humanoid, the method fails to surpass prior works.

---

### Official Review · Reviewer_M8UP · 2023-08-02

**Confidence:** 3
**Originality:** Very Good
**Technical Quality:** Good
**Clarity Of Presentation:** Good
**Impact:** 3

**Recommendation:**

Weak Accept: I recommend accepting the paper, but will not argue for my recommendation if the majority of other reviewers have a different opinion.

**Review:**

Pros:

- This paper presents solid mathematical formulations supported by numerous math proofs in the appendix.

- The authors have invested significant effort in presenting math equations, tables, and figures with a professional and graceful appearance.

Cons:

- Unfortunately, I found it challenging to follow the paper's storyline. In my opinion, the main manuscript lacks self-containment, with insufficient descriptions and explanations. To comprehend the paper, readers are expected to possess the same knowledge and research experience as the authors.

- The main manuscript lacks a connection to related works. While I understand that an 8-page limitation is restrictive for a theory-heavy paper, the absence of a proper discussion of related works makes it difficult to establish a knowledge framework and differentiate this work from previous research. Although the authors include related works in the Appendix, it cannot be assumed that reviewers will read it.

- The paper lacks a clear motivation for this research and its connection to standard Reinforcement Learning (RL). For instance, it would be helpful to explain the difference between using cost and reward in the MDP description and how Cartpole-cost differs from the standard Cartpole task in RL. Additionally, a more detailed description of the costs used in the experiments would be valuable.

- Some of the math definitions and theorems appear to be directly taken from other papers without proper citation. For instance, "Mean Cost Stability" in Definition 2.1 can also be found in Definition 1 of [9], yet the authors state in line 68 that they introduce it as the stability condition in this paper.

- Many of the mathematical formulations in sections 3 and 4 lack proper explanations, making the paper challenging to understand, especially for readers with limited knowledge in the field. For example, from line 86 to line 100 (theorem 3.1), it is unclear what this theorem represents, why it is necessary, and its connection to the formulation in line 98. More reader-friendly explanations are needed to provide insight and intuition, even though the math proofs seem solid.

- Please provide additional descriptions of the experiments, explaining the task settings and the rationale behind their selection in this paper.

- Despite these drawbacks, I believe this paper is founded on solid mathematical principles. However, the missing part blocks conveying these concepts to the readers. Therefore, I suggest the authors consider whether CORL is the appropriate venue for this work, given the heavy emphasis on theory and math proofs in your work. A journal with a main manuscript length of 20+ pages might be a more suitable choice, allowing for more extensive explanations to ensure a technically smooth and coherent narrative.



Critical issue:

- CORL explicitly requires a limitation section in the main manuscript, which this paper failed to include, putting it at risk of desk rejection (https://www.corl2023.org/instructions-for-authors/). I recommend that the authors carefully review the submission guidelines in the future.


$\textbf{Updated review after rebuttal:}$

Most of my concern has been addressed by the authors. I am in favor of the paper now.

**Quality Of The Limitations Section:**

Limitations are addressed clearly

**Questions For Rebuttal:**

Unfortunately, I encountered difficulties in comprehending the paper starting from page three. As such, I anticipate the revised version to include comprehensive explanations for both the mathematical formulations and the experiments.


$\textbf{Updated review after rebuttal:}$

Most of my concern has been addressed by the authors. I am in favor of the paper now.

**Robotics Focus:**

Relevant but unlikely to deploy to hardware in near future

**Summary Of Paper:**

This paper introduces a novel model that addresses sampling-based stability issues by employing Lyapunov stability certification. Additionally, the model combines this approach with the actor-critic algorithm to achieve optimality and stability in various simulated robotics tasks, outperforming the baselines.

**Summary Of Recommendation:**

I cannot recommend this paper as I faced difficulties in understanding the main math definitions and theorems presented in this work. The lack of comprehensive motivation and explanations hinders my ability to grasp the work's contribution and judge its significance.

Nevertheless, it is evident that this paper builds upon a very solid mathematical foundation and includes well-structured proofs. If other reviewers with expertise in control theory and stability analysis hold differing opinions, I would suggest considering their recommendations more seriously than mine.

$\textbf{Updated review after rebuttal:}$

After reading the revised version of the paper, I recommend this paper now. The authors spent a lot of the effort to improve the readability of the paper and allow the researchers in a broad area to understand this work. Combining these updates with the solid math fundamentals, I recommend this work should to be accepted.

---

> ### Author Response · Authors · 2023-08-12
> **To Reviewer M8UP**
>
> Dear Reviewer M8UP,
>
> I hope this message finds you well. As the author/reviewer discussion/rebuttal phase will end on **August 15 at 11:59 PM PT**, we are waiting for your feedback to enhance our paper. We have uploaded the revised version of the paper, incorporating changes to the descriptions of the proposed theorem in Sections 3 and 4. We look forward to your response and greatly appreciate your continued engagement with our work.
>
> Best regards,
>
> Authors

---

### Author Response · Authors · 2023-08-07
**To all reviewers: we have submitted our revised manuscript through the “Rebuttal” button**

Dear reviewers, we would like to thank you for reading and reviewing our manuscript, and your comments did help us improve the manuscript in a more scientific manner. According to your suggestion, we have submitted our revised manuscript through the “Rebuttal” button. We hope that our revisions will help you better understand the paper. We would appreciate it if you can take time to read it carefully. Thank you very much again and we are looking forward to your feedback.

---

### Author Response · Authors · 2023-08-15
**To all reviewers: thank you for you comments and we are looking forward to your feedback**

Dear reviewers, we would like to thank you for reading and reviewing our manuscript, and your comments did help us improve the manuscript in a more scientific manner. Our reply is presented below, and we would appreciate it if you can take time to read it carefully.
Specifically, we have addressed the major concerns of the reviewers by making the following three main changes:

1. We have introduced intuitive descriptions of Theorem 3.1 to make the paper more accessible to readers unfamiliar with the stability field;
2. Comprehensive explanations for the mathematical formulations in Sections 3 and 4 have been added to provide a clearer understanding;
3. We have clarified several unclear parts, such as the connection between the proposed theorem and practical algorithm, experimental settings and the related work about clipping the Lagrange multiplier.

Furthermore, we hope that the revised manuscript can facilitate reviewers to better understand our work. If you have any more questions or need help understanding anything, please feel free to ask.

---

### Decision · Program_Chairs · 2023-08-30

**Decision:**

Accept (Poster)

**Comment:**

The paper presents a rigorous mathematical framework for policy optimization with sampling-based Lyapunov stability. The initial concerns of the reviewers regarding the clarity of the presentation was well addressed by the rebuttal. All reviewers now agree that the paper should be accepted  due to its rigorous mathematical derivation and the convincing results with comparison to relevant baselines. I agree that the paper contains interesting contributions and is of high quality. I therefore recommend acceptance.